

# Assessment of time of emergence of anthropogenic deoxygenation and warming: insights from a CESM simulation from 850 to 2100 CE

Angélique Hameau[1, 2], Juliette Mignot[3], and Fortunat Joos[1, 2]

[1]Climate and Environmental Physics, Physics Institute, University of Bern, Switzerland
[2]Oeschger Centre for Climate Change Research, Bern, Switzerland
[3]LOCEAN Laboratory, Sorbonne Universités, Paris, France

*Correspondence to:* Angélique Hameau (hameau@climate.unibe.ch)

**Abstract.**

Marine deoxygenation and anthropogenic ocean warming are observed and projected to aggravate under continued greenhouse gas emissions. These changes potentially adversely affect the functioning and services of marine ecosystems. A key question is whether marine ecosystems are already or will soon be exposed to environmental conditions not experienced during the last millennium. We find that anthropogenic deoxygenation and warming in the thermocline have today already left the bounds of natural variability in respectively 60 % and 90 % of total ocean area in a forced simulation with the Community Earth System Model (CESM) over the period 850 to 2100. Natural variability is assessed from last millennium (850-1800) results considering forcing from explosive volcanic eruptions, solar irradiance, and greenhouse gases in addition to internal, chaotic variability. Control simulations are typically used to estimate variability. However, natural variability in oxygen ($O_2$) and temperature (T) are systematically larger than internal variability (e.g. the latter amounts to 20 % for T and 60 % for $O_2$ in the thermocline), rendering such estimates of natural variability to be biased low. Results suggest that anthropogenic change in apparent oxygen utilisation (AOU) and in $O_2$ solubility ($O_{2,sol}$) are earlier detectable by measurements than in $O_2$ in the tropical thermocline, where biological and solubility-driven $O_2$ changes counteract each other. Both natural variability and change in AOU are predominantly driven by variations in circulation with a smaller role for productivity. Ventilation becomes more vigorous in the tropical thermocline by the end of the $21^{st}$ century, whereas ideal age in deep waters increases by more than 200 years until 2100. Different methodological choices are compared and the time for an anthropogenic signal to emergence (ToE) is earlier in many thermocline regions when using variability from a short period, the control, or estimates from the industrial period instead variability from the last millennium. Our results highlight that published methods lead to deviations in ToE estimates, calling for a careful quantification of variability and that realised anthropogenic change exceeds natural variations in many regions.

*Copyright statement.* TEXT



## 1 Introduction

Oceanic oxygen $O_2$ concentrations have been observed to decrease over the past 50 years (e.g. Stramma et al., 2008; Helm et al., 2011; Talley et al., 2016; Ito et al., 2017; Schmidtko et al., 2017) and are projected to further decline under anthropogenic climate change (Sarmiento et al., 1998; Plattner et al., 2001; Bopp et al., 2002; Keeling and Garcia, 2002; Schmittner et al., 2008; Frölicher et al., 2009; Shaffer et al., 2009; Cocco et al., 2013; Bopp et al., 2013; Bopp et al., 2017; Niemeyer et al., 2017; Battaglia and Joos, 2018; Fu et al., 2018). This deoxygenation, in combination with other stressors (Bopp et al., 2013; Cocco et al., 2013) such as ocean warming (IPCC, 2013), acidification (Orr et al., 2005) and declining primary productivity by phytoplankton (Steinacher et al., 2010; Laufkötter et al., 2015), poses major risks for the function of marine ecosystems and the services they provide (Pörtner et al., 2014; Gattuso et al., 2015; Deutsch et al., 2015; Mislan et al., 2017; Breitburg et al., 2018).

The modern marine $O_2$ cycle is characterised by its low $O_2$ inventory. Only about half a percent of the freely available $O_2$ is stored in the ocean and waters are anoxic in tropical eastern boundary upwelling systems (Bianchi et al., 2012; Brandt et al., 2015). Surface $O_2$ concentrations are close to equilibrium with the atmosphere driven by a fast air-sea gas transfer (Gruber et al., 2001). Cold high latitude surface waters are characterised by high solubility and high $O_2$ compared to warm tropical and subtropical surface waters. Within the ocean, water is primarily consumed by aerobic remineralisation (Emerson and Bushinsky, 2014), roughly balancing net $O_2$ production in the surface. The $O_2$ concentrations in subsurface waters reflect a balance between physical $O_2$ supply by advection, mixing, and convection and biological consumption of organic matter exported to the deep (Frölicher et al., 2009; Gnanadesikan et al., 2013; Duteil et al., 2014; Keller et al., 2016; Azhar et al., 2017). Ongoing anthropogenic warming of the upper ocean leads to a solubility-driven decrease in $O_2$. It also leads to an increased stratification (Manabe et al., 1991). In turn, mixing and overturning generally slows and $O_2$ supply rates to the deep decrease, while more time is left for $O_2$ consumption in subsurface waters. This causes a corresponding $O_2$ decrease, though specific model responses in circulation, biological organic matter export and $O_2$ consumption and in $O_2$ concentrations are complex in space and time and regionally uncertain (e.g., Frölicher et al., 2009; Long et al., 2016; Azhar et al., 2017; Oschlies et al., 2017; Palter et al., 2018; Battaglia and Joos, 2018).

While it is expected that continued anthropogenic emissions of $CO_2$ and other greenhouse gases cause large scale deoxygenation, it is less clear whether and to which extent $O_2$ concentrations in the ocean have already left the bounds of natural variability of the recent millennia. Many ecological and socio-economic systems are adapted to the level of natural variability of the recent past. Therefore, forcing environmental conditions outside the bounds of recent natural variability may trigger adverse impacts (Pörtner et al., 2014). It is still an open question to which extent marine species and ecosystems are already confronted with "unfamiliar" environmental and climatic conditions not encountered during the last millennium. There is in particular a lack of proxy reconstructions documenting how $O_2$ might have evolved during the current interglacial period and a lack of modelling studies addressing natural variability of $O_2$ during the recent millennium, while observational information, though limited in space and time, is available on $O_2$ variability in the modern ocean (Mecking et al., 2008; Brandt et al., 2015; Long et al., 2016; Ito et al., 2017).



The time of emergence (ToE) is an established metric to estimate when a forced anthropogenic change or "signal" leaves the bounds of variability or "noise" (Hawkins and Sutton, 2012). ToE is the point in time when the signal becomes larger than the noise of variability. Thus, the magnitude of the anthropogenic signal is compared to the magnitude of the noise to compute ToE. Different attempts to detect and attribute $O_2$ change are using optimal fingerprinting methods and consider

spatial patterns of change (Andrews et al., 2013; Long et al., 2016). ToE has been estimated for physical climate variables (e.g. Mahlstein et al., 2013; Hawkins et al., 2014; Frame et al., 2017), land carbon stocks and fluxes (Lombardozzi et al., 2014) and marine properties including ocean acidification impacts and alkalinity (Ilyina et al., 2009; Friedrich et al., 2012; Ilyina and Zeebe, 2012; Hauri et al., 2013), sea surface temperature, pH, $CO_2$ partial pressure and dissolved inorganic carbon (Keller et al., 2014), marine biological productivity, biological export fluxes, surface chlorophyll and surface nitrate (Henson et al., 2016)

and air-to-sea carbon fluxes (McKinley et al., 2016). A limited number of studies also addressed the ToE for $O_2$ concentrations in the thermocline, often in combination for other tracers (Rodgers et al., 2015; Frölicher et al., 2016; Henson et al., 2016; Long et al., 2016; Henson et al., 2017). Most of these ToE studies are directed towards the detection of the current anthropogenic trend by using modern measurements systems (Rodgers et al., 2015; Frölicher et al., 2016; Henson et al., 2016; Long et al., 2016). Thus, the question is how many years of modern measurements, starting in 1956 or later, are needed to detect the

anthropogenic $O_2$ signal. Alternatively, (Henson et al., 2017) address the question when ecosystems are exposed to conditions outside the range of previously experienced seasonal variability and, hence, ToE is estimated relative to preindustrial.

Published ToE studies employ a wide range of different methods. For example, the anthropogenic signal is computed as a linear trend over a few decades (Keller et al., 2014; Rodgers et al., 2015) or a linear trend over the industrial period and the $21^{st}$ century (Henson et al., 2017) or by a polynomial fit to simulated data (Carter et al., 2016). Even a wider range of approaches is

used to estimate variability or noise. Noise is taken as the standard deviation (STD) of annual extrema from a control simulation (Henson et al., 2017) or STD remaining after removing anthropogenic trends (Keller et al., 2014; Henson et al., 2016) or STD from a model ensemble for the 1920-1950 period (Long et al., 2016), by appropriately adding the STD of annual values plus monthly values plus from measurement uncertainty (Carter et al., 2016), the amplitude of the simulated preindustrial annual cycle (Friedrich et al., 2012), the extrema simulated over the historical period (Mora et al., 2013), the standard deviation in

the projected trends from model ensembles (Rodgers et al., 2015; Frölicher et al., 2016). A characteristic of these different estimates of variability is that variability is generally estimated for a limited temporal period and, perhaps, more important that variations from natural forcing variations are considered not at all (e.g. Henson et al., 2017) or to a limited extent. For example, forcing from explosive volcanic eruptions (Sigl et al., 2015) and solar irradiance variations (Muscheler et al., 2007) are by design not included in model control simulations. These shortcomings may bias such estimates of natural variability

systematically low. As a consequence, the time when the anthropogenic signal leaves the bounds of natural variations would also be biased towards early emergence.

Earth system responses to natural variations in external forcing factors contribute to total natural variability in climate and biogeochemical variables. These externally-forced variability is additional to the partly chaotic internal variability of the climate system. Last millennium climate reconstructions (PAGES 2k Consortium et al., 2015; McGregor et al., 2015; PAGES2k

Consortium et al., 2017), last millennium climate simulations (Crowley, 2000; Ammann et al., 2007; Fernández-Donado et al.,




2013; Camenisch et al., 2016), and the few existing last millennium earth system model (ESM) simulations with enabled carbon and biogeochemical cycles (Lehner et al., 2015; Jungclaus et al., 2010) and studies using their output (Brovkin et al., 2010; Keller et al., 2015; Chikamoto et al., 2016a) all suggest substantial variations in physical and biogeochemical variables in response to the volcanic eruptions and solar irradiance variations (Schmidt et al., 2011; Jungclaus et al., 2017) of the

last millennium. Regarding biogeochemical cycles, a substantial role of natural forced variability is also found in factorial simulations with and without volcanic forcing (Frölicher et al., 2009; Frölicher et al., 2011; Frölicher et al., 2013). Frölicher et al. (2009) document in their ESM ensemble that volcanic eruptions cause significant interannual and decadal variability in $O_2$, that volcanic perturbations in oceanic $O_2$ gradually penetrate the ocean's top 500 m and persist for several years, and that these forced variations are additional to modelled and observed $O_2$ variability associated with the North Atlantic and Pacific

Decadal Oscillation. In conclusion, the available evidence suggest that forced natural variability should not be ignored when comparing the relative importance of anthropogenic trends versus natural variability.

The goals of this study are to quantify when the anthropogenic marine deoxygenation and warming leaves the bound of natural variability of the last millennium and to estimate the relative role of natural forced and internal variability in marine $O_2$ and temperature variations. A further goal is to document the influence of different methodological choices on estimates of

ToE. We exploit results from one of the few available ESM simulations with active carbon and $O_2$ cycle that covers the entire last millennium, the industrial period and the $21^{st}$ century (Lehner et al., 2015). We further document the variability in terms of standard deviation of apparent oxygen utilisation, the solubility driven $O_2$ change and their covariance as well as variability in ideal age and export production of particulate organic carbon and the anthropogenic change in these variables. This allows us to discuss the role of solubility, biological export, and circulation for $O_2$ variability and anthropogenic signal.

## 2    Method

### 2.1    Model and simulations

#### 2.1.1    Model description

The model used for this study is the Community Earth System Model version 1.0 (CESM1), released in 2010. It is a fully coupled state-of-the-art Earth system model (Hurrell et al., 2013). This version of the model was used in the Coupled Model

Intercomparison Project (CMIP5). Its physics relie on the Community Climate System Model (CCSM4; Gent et al., 2011), which includes modules for the atmosphere, the land, the sea-ice and the ocean, all coupled by a flux coupler. As compared with CCSM4, CESM1 additionally includes a fully interactive carbon cycle between the atmosphere, ocean and land modules.

The atmospheric module is the Community Atmosphere Model (CAM4; Neale et al., 2010), with a horizontal resolution of 1.25° x 0.9° and 26 vertical levels. CAM4 provides an interactive coupled biogeochemistry module (CAM-chem). The

land module is the Community Land Model (CLM4; Lawrence et al., 2012). It operates on the same horizontal grid as the atmospheric component. The land surface is represented as a hierarchy of subgrid types, including glacier, lake, wetland, urban and vegetated land units. The ocean is simulated by POP2 (Parallel Ocean Program version 2; Smith et al., 2010; Danabasoglu




et al., 2011), with 60 vertical levels. The horizontal resolution varies around 1°. It is higher at low latitudes (around the equator) and around Greenland to where the North Pole is displaced in order to avoid singularity problems in the ocean model equations. Note that for convenience, the global maps shown here are re-gridded using a regular grid. The Community Ice Code (CICE4; Hunke et al., 2010) stands for the sea ice component. It operates on the same horizontal resolution as the ocean module.

The Biogeochemical Elemental Cycling model (BEC; Moore et al., 2002 and 2004) is implemented in POP2. It is built on traditional phytoplankton-zooplankton-detritus food web models (Doney et al., 2009). It carries three different phytoplankton types: diatoms, diazotrophs and small phytoplankton. The photosynthesis and associated production rate of oxygen depend on the phytoplankton type, the solar irradiance and nutrient-limitation terms (Cullen, 1990). The nutrient-limitation terms are scaled by Redfield ratios, which represent the nutrient assimilation type; C:O varies depending whether $NH_3$ or $NH_4^+$ is

assimilated or N-fixed (Nitrogen fixation; only for diazotrophs). At the surface, the rate of air-sea oxygen transfer depends on the modelled wind speed, the temperature-dependent Schmidt number (Wanninkhof Rik, 1992) and the air-sea partial pressure difference in $O_2$. The solubility component of $O_2$ ($O_{2,sol}$) is parametrised as function of temperature (T) and salinity (S) (Garcia and Gordon, 1992). It is defined as the $O_2$ concentration in equilibrium with an atmosphere of standard composition, fully saturated with water and with a total pressure of one atm. The oxygen content is homogenised within the mixed layer.

The typical time scale to equilibrate the oxygen concentration in the mixed layer with the atmosphere by gas exchange is about one month.

The modelled consumption of oxygen occurs through remineralisation of organic material, respiration by zooplankton, and grazing of the phytoplankton. After the death of the phytoplankton, the transformed biomass is distributed among dissolved, particulate and inorganic pools. The distribution varies by type of plankton and type of mortality (Moore et al., 2004). Aggre-

gated biomass is routed to the sinking Particulate Organic Matter (POM) pools and sinks at a rate of 20.0 m day$^{-1}$. Carbon export and remineralisation are following Armstrong et al. (2001). Remineralisation of the detrital pools is parameterised with the condition that $[O_2] >= 4$ mmol m$^{-3}$ and a temperature-dependent function. The length scale of remineralisation for the sinking POM pool varies from 200 to 1000 m (Moore et al., 2002). Organic material reaching the ocean floor is remineralised instantaneously, i.e., no sediment module is included.

The global circulation model (POP2) also includes a water age tracer (ideal age). It is set to zero at the surface and ages every day in the ocean interior.

### 2.1.2 Description of the simulations

This study uses results from a forced, transient simulation spanning from 850 to 2100 CE (Common Era) and a corresponding control simulation, both performed with CESM1.0.1. The experimental set-up is described in detail by Lehner et al. (2015) and

further results of these runs are described by Bothe et al. (2015), Keller et al. (2015), Camenisch et al. (2016) and Chikamoto et al. (2016b). The reference simulation (CTRL) was branched from the CMIP5 CCSM4 (Gent et al., 2011) pre-industrial control and run for 258 years with the 850 CE external forcing set to allow a spin up. At nominal year 850 CE, the forced simulation was branched off. The CTRL was continued for another 518 years from 850 to 1368 CE with unchanged forcing. The transient forcing largely follows the protocol of the Paleoclimate and Modelling Intercomparison Project 3 (Schmidt et al.,



2011), applying reconstructed variations of the volcanic forcing (Gao et al., 2008), land use changes (Pongratz et al., 2008; Hurtt et al., 2011), and fossil fuel emissions (1750 to 2005 CE, following Andres et al., 2012). The total solar irradiance is taken from the reconstruction by Vieira and Solanki (2010) but scaled to have an amplitude change from the Maunder Minimum to present day of 0.2 % rather than 0.1 %, consistent with Bard et al. (2000). Over the period 2005–2100 CE, the representative

concentration pathway RCP8.5 (Moss et al., 2010) is used (see Fig. A1c for an overview of the forcings).

### 2.1.3 Model evaluation

We briefly compare model results for $O_2$ (average between 1986 and 2005) to observations (Garcia et al., 2013). Earlier studies, Doney et al. (2006) and Najjar et al. (2007), have described strengths and weaknesses of the climate model CESM. And Lehner et al. (2015) discuss also the simulations analysed here.

In the thermocline (here defined as the layer 200-600 m), the model reproduces the main features of the $O_2$ distribution given by the World Ocean Atlas 2013 (Garcia et al., 2013) as illustrated in Fig. A2a. $O_2$ concentrations are generally high at these intermediate depths in the mid- and high-latitudes of the Southern Hemisphere as well as in the mid-latitudes of the Pacific and northern Atlantic. Both the model and the World Ocean Atlas show low concentrations in the equatorial thermocline and in the northern North Pacific thermocline (Moore et al., 2013). The model simulates too widely expanded Oxygen Minimum Zones

(OMZs, defined here as areas where the oxygen concentration is below $20\,\mathrm{mmol\,m^{-3}}$; magenta contours) in the eastern Pacific, Atlantic and Indian oceans. Similar biases have been identified in other models and attributed to biases in the production and remineralisation of particulate organic matter and to deficiencies in the representation of the Equatorial Undercurrent in Earth system models (Bopp et al., 2013; Cocco et al., 2013; Brandt et al., 2015; Cabré et al., 2015; Oschlies et al., 2017).

Regarding the temperature mean distribution (Fig. A2d), the model is able to simulate the isopycnal structure at intermediate

depths reported by Locarnini et al. (2013): cold water masses at the poles, the centre of the subtropical gyres show higher temperature than the surroundings (Fig. A2b). Nevertheless the model simulates colder water in the Southern Ocean and in the equatorial Pacific band and warmer water in the eastern North Atlantic (isotherm 14° C) in the thermocline compared to the observations.

## 2.2 Analysis tools

### 25 2.2.1 Correcting for model drift and removing millennial trends

At year 850 CE of the CTRL simulation, the upper 500 meters of the ocean are generally equilibrated with the forcing resulting in negligible linear drifts of T ($-2.8 \, 10^{-2}$ °C per century), S ($-5.0 \, 10^{-3}$ permil per century), $O_2$ ($0.22\,\mathrm{mmol\,m^{-3}}$ per century) and other properties in the CTRL (not shown). The drift increases with depth. It starts to become noticeable compared to the variability around 500 m depth. In this study, we focus on variability within the thermocline (200-600 m) where drifts are still

small and do not affect conclusions. Nevertheless, we use the results from the CTRL to estimate and correct for model drifts in all the studied variables. Because the control simulation drift seems to diminish by the end of the period, an exponential curve $(a(1 - e^{-|b|t}) + c)$ was fitted to the annual outputs of the CTRL simulation at each grid cell and for each variable of interest,




and extended to 2100 CE. The fit was then removed from the original output of the CTRL and forced simulations (Fig. A1a and b; solid purple curve). This results in the removal of any long-term trend in the CTRL and to a large extent in the surface ocean and the upper thermocline in the forced run.

Yet, the drift-corrected forced last-millennium simulation shows residual millennial-scale trends in the subsurface waters and the deep ocean. We do not exactly know the origin of the millennial-scale trends in the deep ocean, but hypothesise that these trends are a response to including volcanic forcing in the forced simulations, whereas volcanic forcing is absent in the CTRL and spin up. This leads on average to a negative radiative forcing compared to the spin up and control simulation (Gregory, 2010). The deep ocean only slowly adjusts to this averaged negative forcing possibly leading to long-term cooling and a corresponding increase in $O_{2,sol}$ and $O_2$ in the deep ocean. In this study, we are primarily interested in investigating

variability on time scales ranging from years to many centuries in the upper ocean and to detect anthropogenic trends from the background of natural interannual to centennial scale climate variations as representative for the last thousand years. Slow trends influence the computation of metrics such as the standard deviation around a mean value and Time of Emergence (ToE). Although millennial scale trends are small in the surface ocean and the thermocline, we fitted a linear trend to the model output over the period 850-1800 CE at each grid cell and for each variable of interest. This trend is then removed from the forced

simulation to exclude millennial-scale variability (Fig. A1a; red solid curve). Figure A1a and b illustrate the computation of drift- and trend-corrected fields from the original outputs for the globally averaged ocean and the thermocline (200-600 m). We note that these corrections have a small influence on the results in the upper ocean, our main study area, and do not affect our main conclusions.

### 2.2.2 Separation of $O_2$ concentration into components

Marine $O_2$ concentration can be expressed as the sum of two components: $O_2$ solubility ($O_{2,sol}$) and Apparent Oxygen Utilisation (AOU) following earlier studies (e.g. Frölicher et al., 2009; Bopp et al., 2017; Ito et al., 2017). $O_{2,sol}$ is approximated by the saturation concentration as described in Sect. 2.1.1. It depends non-linearly on T and S, but the variations in $O_{2,sol}$ are mainly driven by the variations in temperature. AOU is computed as a residual from modelled $O_2$ and diagnosed $O_{2,sol}$:

$$[O_2] = [O_{2,sol}] - AOU \tag{1}$$

AOU predominantly reflects $O_2$ respiration due to remineralisation of organic material in the model. It depends on the amount of organic matter sinking expressed in this study by the Particulate Organic Carbon production (POC production) and on water mass age (ideal tracer) as dictated by circulation, mixing and convection. For completeness, we note that the diagnosed AOU component is additionally influenced by deficiencies in the saturation concentration $O_{2,sol}$ to represent the solubility component. These arise due to mixing of different source waters given the non-linear relationship between solubility

and T, S as well as by incomplete air-sea surface equilibration of these source waters.



### 2.2.3 Estimation of the Time of Emergence

In order to detect the changes due to anthropogenic forcings, we use the Time of Emergence (ToE) concept (Hawkins and Sutton, 2012). We estimate the ToE when decadal-scale changes due to anthropogenic forcing in $O_2$, temperature and related variables emerge from natural variations (Eq. 2). We express drift- and trend-corrected concentrations as annual anomalies

relative to a preindustrial reference period spanning from 1720 to 1800 CE (Hawkins et al., 2017). The natural variability or noise, $N$, is computed as one standard deviation of the anomalies over the period from 850 to 1800 CE in the forced simulation and for each water volume (grid cell to global ocean) and variable of interest. Note that considering spatially-averaged variables, their standard deviation is computed from spatially averaged values, rather than by taking the averaged deviations of the corresponding individual grid cells. The low frequency climate change, $S$, is diagnosed as the spline-fitted

anomalies using a cut-off period of 40 years (Enting, 1987) to remove short-term variations. The ToE is determined as the time when the signal $S$ becomes for the first time larger than twice the noise $N$ (Fig. A3).

$$ToE : \frac{S}{N} \geqslant 2 \tag{2}$$

## 3 Results

### 3.1 Evolution for globally-averaged perturbations in ocean temperature and oxygen

We start the presentation of our results by first discussing variability and trends for averaged temperature (T) and dissolved oxygen ($O_2$) for the surface, the thermocline (200-600 m) and the whole ocean. The magnitude of variability and anthropogenic trends is larger for the surface ocean and the thermocline than for the deep ocean (Fig. 1).

Globally-averaged T and $O_2$ show near exponential perturbations at all depths (Fig. 1, right) during the industrial period and the $21^{st}$ century in response to the prescribed anthropogenic forcing. Globally-averaged T increases by 3.7, 2.0, and 0.7 °C

from the preindustrial reference period (1720-1800 CE) until 2100 in the surface ocean, the thermocline and the whole ocean. For comparison, the mean surface air temperature increase is 5.4 °C by 2100. The ocean mean warming is about a factor of five lower than the global-mean surface ocean warming. Regarding $O_2$, the anthropogenic perturbation leads to a $O_2$ decrease by about 15 mmol m$^{-3}$ (-6 %) in the spatially-averaged surface ocean, by 16 mmol m$^{-3}$ (-11 %) in the thermocline and by 10.5 mmol m$^{-3}$ (-5 %) when averaged over the whole ocean. The anthropogenic trends are qualitatively consistent with

earlier observational (Keeling and Garcia, 2002; Keeling et al., 2010; Bakun, 2017; Ito et al., 2017; Schmidtko et al., 2017) and modelling studies (Frölicher et al., 2009; Bopp et al., 2013; Cocco et al., 2013; IPCC, 2013; Bopp et al., 2017).

Last millennium variability in averaged surface ocean T and $O_2$ appears to be dominated by interannual to decadal variability, whereas large variations on multi-decadal and centennial time scales are simulated for the thermocline and the whole ocean (Fig. 1, left). During the period 850 to 1800 CE, simulated global mean sea surface temperature (SST) varies generally within

an interval of about ±0.3 °C and global mean surface $O_2$ within ±1.2 mmol m$^{-3}$ relative to the reference period (1720-1800 CE). Large global mean SST changes of up to 2 °C cooling are modelled after large explosive volcanic eruptions. These





are accompanied by large positive anomalies in surface mean $O_2$ of up to 7 mmol m$^{-3}$. The large post-eruption anomalies decay within a few years to decades in the surface ocean. In the averaged thermocline, annual T varies within -0.15 °C and +0.1 °C and $O_2$ between –0.7 and 4 mmol m$^{-3}$ relative to the reference period. These variations occur on multi-decadal-to-centennial time scales. The imprint of large explosive eruptions are visible as abrupt, sudden perturbations (e.g. at the year

1258), followed by long-term shrinking of these initial perturbations. Variability for the whole ocean shows a similar variations as in the thermocline, but with an order of magnitude smaller peak variations.

### 3.2 ToE, natural variability and anthropogenic change in the thermocline

#### 3.2.1 ToE

Figures 2a and b show the ToE spatial patterns for $O_2$ and T in the thermocline (200-600 m). Here the ToE is indicative for the

emergence of a signal on the horizontal scale of a grid cell and vertically-averaged between 200 and 600 m. These figures show well defined patterns, with zones of early emergence (before 2020) and zone of late (2020 < ToE < 2099) or no emergence (by 2099) of the anthropogenic signal.

In general, the human-induced $O_2$ changes in the thermocline emerge early (before 2020) in high- and mid-latitudes, whereas they emerge late or not at all in the tropics (Fig. 2a). Nevertheless, late or no emergence until 2100 is found in the subtropical

Atlantic, western South Pacific and Indian ocean, while a rather early emergence is simulated in the eastern equatorial Atlantic and the Indian ocean subtropical gyre.

In the case of temperature (Fig. 2b), the anthropogenic signal emerges generally before the end of the 21$^{st}$ century in the thermocline, except in small areas in the North Atlantic and in the western North Pacific. In contrast to $O_2$, early emergences of the anthropogenic warming are simulated in the subtropical Atlantic gyres, and late emergences are simulated in the subpolar

North Atlantic and of the Pacific gyre regions. The anthropogenic T signal emerges early in the equatorial regions, again in contrast to $O_2$.

Interestingly, the spatial patterns of ToE for $O_2$ and T seem generally inversely related in the thermocline. As a result, the spatial pattern of the difference between ToE of T minus ToE of $O_2$ resembles the spatial pattern of ToE of $O_2$ (Fig. 2c). In large regions, particularly in the tropics, the mid-latitude regions of the Atlantic and along the coast of South America, the

anthropogenic T signal emerges much earlier than the $O_2$ signal in the thermocline (brown areas; Fig. 2c). Because ocean temperature and ocean physics influence marine biogeochemical cycles and $O_2$, one may expect T changes to emerge before $O_2$ changes. Yet in some areas, such as the subtropical Pacific gyres, the Southern Ocean, the northern equatorial Atlantic, and the North Atlantic subpolar gyre the anthropogenic signal emerges first in $O_2$. The reasons for these results are analysed in Sect. 3.4. The following section is first dedicated to a more in-depth analysis of the signal and the noise that both define ToE.

#### 3.2.2 Natural variability and anthropogenic changes

Considering that ToE is a signal-to-noise problem, we compare the magnitude of the signal (anthropogenic changes of $O_2$ and T; Fig. 3c, d) as well as of the noise (the amplitude of the natural variability; Fig. 3a, b) to understand why a signal emerges




early or late. Natural variability of $O_2$ in the thermocline (Fig. 3a) range from less than 1 mmol m$^{-3}$ to more than 10 mmol m$^{-3}$ (STD ±2.50 mmol m$^{-3}$; 850 to 1800 CE). Variability is small in the core of the $O_2$ minimum zones in the tropical Indian ocean and in the eastern tropical Pacific and Atlantic as $O_2$ remains depleted. Variability is generally high at the edge of the major oceanic gyres, including transition zones to $O_2$ minimum regions.

The anthropogenic signal in $O_2$ remains relatively small in the $O_2$ minimum zones and the subtropical Atlantic gyres (Fig. 3b). $O_2$ is projected to increase in the thermocline in the southern subtropical Indian gyre region and in the tropical Pacific, whereas $O_2$ is projected to decrease in mid- and high-latitudes in the thermocline. The largest decrease is found in the North Pacific, up to 50 mmol m$^{-3}$ by the end of the $21^{st}$ century.

By definition, the areas of early emergence result from a high signal-to-noise ratio. A local signal may emerge early compared
to other regions due to a relatively low variability or a relatively high anthropogenic signal, or a combination of both. For example, simulated $O_2$ shows generally a relatively weak natural variability (± 2 mmol m$^{-3}$; Fig. 3a) in the thermocline south of 30° S and a large anthropogenic $O_2$ change (>+12 mmol m$^{-3}$ by 2100 CE; Fig. 3c). In the north Pacific, however, the standard deviation is high (± 10 mmol m$^{-3}$), but the anthropogenic signal is very large as well (-50 mmol m$^{-3}$ by 2100 CE). In the eastern tropical Atlantic, the $O_2$ signal is weak, but the natural variability is even weaker. In these three cited regions,
anthropogenic changes emerge relatively early, but for different reasons.

On the contrary, a low signal-to-noise ratio will induce a late or no emergence of the anthropogenic signal. The anthropogenic $O_2$ signal has not emerged by 2100 in the thermocline in the western tropical Pacific, the western coastal Indian and the subtropical Atlantic gyres because of a strong natural variability (±10 mmol m$^{-3}$) and relatively weak changes in $O_2$ (- 6 mmol m$^{-3}$ by 2100 CE). But in the southern tropical Indian Ocean, North Atlantic subpolar gyre and the eastern tropical
Pacific, the $O_2$ changes outweigh the natural variability leading to emergence by the end of the $21^{st}$ century.

In general, the temperature signal-to-noise ratio is high in the thermocline, and the emergence of human-induced changes occurs before the end of the $21^{st}$ century. The thermocline temperature varies naturally less than ±1 °C (Fig. 3b) and the anthropogenic changes are between +1 °C and +4 °C by the end of the $21^{st}$ century (Fig. 3d). However, the subtropical Pacific gyres, the northern tropical Atlantic and subtropical Indian gyre show slightly more intense natural variations which delay the
emergence of the anthropogenic signal. In parts of the western North Pacific and the North Atlantic, temperature variability in the thermocline is high and the anthropogenic changes remain within the range of natural variability.

### 3.3    Sensitivity of ToE to methodological differences

Different methodological choices were applied in earlier studies to estimate ToE for precipitation (Giorgi and Bi, 2009), air surface temperature (Karoly and Wu, 2005; Diffenbaugh and Scherer, 2011; Hawkins and Sutton, 2012), SST, pCO$_2$, pH,
dissolved inorganic carbon (Keller et al., 2014), primary production or $O_2$ (Rodgers et al., 2015; Carter et al., 2016; Frölicher et al., 2016; Henson et al., 2016; Long et al., 2016; Henson et al., 2017). Different definitions and methods are used to estimate the noise (or natural variability), the anthropogenic signal and the ToE. There seems to be no consensus on the method to estimate ToE. In the following part, in order to gain confidence in the ToE estimates presented above, the influence of different choices for the estimate on ToE is investigated.



### 3.3.1 Noise estimated from internal variability of a control simulation

A prevailing way for estimating the background noise is to consider the internal variability, using the temporal standard deviation (STD) of the control simulation of the grid point or the averaged domain (Hawkins and Sutton, 2012; Maraun, 2013; Long et al., 2016; Henson et al., 2017). We defined the total natural variability as the combination of the internal variability

and the naturally forced variability. The comparing the internal and total natural $O_2$ variations, by using STD of the CTRL and forced simulation (LM) as metric shows that the external natural forcing enlarges the estimated natural $O_2$ variability (Table 1). Indeed, the internal variability represents only 22 % of the total natural variability in the global $O_2$ inventory, 61 % in the $O_2$ inventory of the thermocline, and 47 % in the global mean surface $O_2$ concentration. Moreover, Frölicher et al. (2009) show that explosive volcanic eruptions influence marine $O_2$ for several years in the top 500 m, in accordance to Fig. 1. Therefore,

in the context of detection of the anthropogenic changes, considering only the internal variability seems to underestimate the range of natural variability arising from solar irradiance changes and volcanic eruptions, and leads to earlier emergence of anthropogenic changes on the global scale.

Figure 4 compares the externally forced natural variability with the internal variability in each grid cell in the surface layer, in the thermocline (200-600 m), and in the entire water column for $O_2$ and T, again using STD as a metric. There are many

regions where the ratio between the STD of the forced versus those from the CTRL simulation is close to one indicating that internal and total natural variability are approximately equal (Fig. 4). In particular, total and internal natural variability in $O_2$ is comparable in most thermocline regions (Fig. 4c). However, there are also large regions where the ratio of total to internal variability in $O_2$ and in T is substantially larger than one. Such regions include the tropical and mid-latitude Atlantic, the Arctic and a belt around Antarctica when considering the entire water column (Fig. 4e, f). Total natural variability is up to a factor

of two or more larger than internal variability in T in the thermocline of the tropical and mid-latitude Atlantic, in the Arctic, in parts of the Southern Ocean and of the eastern tropical and subtropical Pacific (Fig. 4d). Surprisingly, the SST shows more variability at high-latitudes in the control simulation rather than in the forced simulation (Fig. 4f). The exact reasons for these differences are beyond the scope of the present study. But we hypothesise that the radiative cooling driven by explosive volcanic eruptions may decrease SST in high latitudes, leading to a more extended sea ice, and therefore reducing SST variability. Yet

in general, variability in SST is in general larger in LM than in CTRL in low and mid latitudes.

Total natural variability is not only substantially lager than internal variability for the vertically-integrated water column and thermocline but also on the level of the individual grid cell in the surface layer, in particular in tropical regions (Fig. 4a, b). In other words, forced natural variability is not only important when integrating its imprint over large volumes, but may also be of importance on the local scale.

In conclusion, using results from a control simulation to estimate natural variability leads to an underestimation of total natural variability in specific regions with corresponding consequences on the estimations of ToE as illustrated by Fig. 5. Nevertheless, the results from a control simulation appear to yield a reasonable approximation of simulated natural variability in $O_2$ and T on the local scale in the thermocline. Regions with larger differences are located at the edges of the subtropical gyres in the North Pacific and the tropical Atlantic (Fig. 5a, b).





### 3.3.2 Noise estimated from the period 1720-1800

Estimates of noise may further be sensitive to the choice of period. Hawkins et al. (2017) define the time period 1720-1800 CE as the optimal pre-industrial period. This pre-industrial period is justified by rather normal conditions during 80 years: relatively stable TSI and small volcanic eruptions (Fig. A1c). Because fully coupled millennial scale simulation are expensive and relatively rare, we compare the STD of T and $O_2$ in the thermocline in the forced LM simulation for the period 850-1800 versus the shorter pre-industrial reference period 1720-1800 CE (Fig. 6a, b).

In a large part of the thermocline, STD in T and $O_2$ are similar (within $\pm 10$ %), reflecting a similar estimated natural variability. Differences in STD are for example found in regions of the South Pacific, the eastern North Pacific, the Arctic Ocean and for $O_2$ in the Arabian Sea. The resulting ToE are compared to the one using STD from the period 850 to 1800 CE (Fig. 6c, d). For $O_2$, substantially earlier ToE are estimated in large parts of the Pacific, the Arctic and the Southern Ocean when using the PI-period to approximate the natural variability. However changes in oxygen are estimated to appear later in some parts of the equatorial Atlantic, in the Arabian Sea and in a few grid cells in the Pacific and Arctic. Similarly, earlier ToE for T are found for example in large parts of the Pacific when using the variability from the PI period. The results suggests that a century-long period of the forced simulation may not yield robust results for variability and ToE in all regions.

### 3.3.3 Noise estimated from a simulation over the industrial period

Most available carbon-cycle enabled ESM simulations addressing anthropogenic change begin in the $19^{th}$ century. Then, the total natural variability cannot be directly assessed and the internal variability can only be determined if the corresponding control simulation has been also produced. Instead, Keller et al. (2014) and Henson et al. (2016) used the standard deviation of the residual variability to estimate natural variability. The residual variability is defined as the difference between the full signal and a low-pass filtered signal designed to extract the effect of the anthropogenic forcing. Here, a spline with a cut-off period of 40 years is used to compute the filtered signal from the year 1800. The standard deviation of the residual variability is computed for the period 1850 to 2005 CE.

Surprisingly, STD of $O_2$ and T in the thermocline as computed from the residual variability of the 1850 to 2005 CE period is much smaller than as computed from the Last Millennium (LM) simulation (850 to 1800 CE) in many regions (Fig. 6e, f). Particularly large differences between the estimates of STD (100 % or more) are found in the Atlantic and high latitude regions for T and $O_2$ and in large parts of the Pacific for $O_2$.

The stronger variability in LM compared to the residual variability likely results from low frequency variability included in LM but removed by the spline in the residuals. The residual variability included only the variability that does not pass the low pass spline filter. Hence, it does not include periods much larger than the cut off period of the spline, taken here to be 40 years. As a consequence of the smaller variability, ToE for $O_2$ and T in the thermocline is earlier when using STD from the residuals instead of STD from the 850 to 1800 CE period to compute ToE (Fig. 6g, h).



### 3.4  Apparent oxygen utilisation, $O_2$ solubility, ventilation, and organic matter cycling

In order to gain insight on the processes underlying $O_2$ variations, we analyse the variability, anthropogenic induced change, and ToE of the apparent oxygen utilisation (AOU) and of the $O_2$ solubility component ($O_{2,sol}$). $O_{2,sol}$ variations are driven by SST changes and AOU variations mainly reflect the imprint of changes in water mass ventilation and in the remineralisation

of organic matter. Here, ventilation is diagnosed by an ideal age tracer and changes in remineralisation of organic matter are linked to the production of organic matter.

### 3.4.1  Natural variability

Changes in AOU largely explain most of the preindustrial variations in $O_2$ with a generally small role for $O_{2,sol}$ both at the global scale and on average in the thermocline (Fig. 1a, b). The $O_{2,sol}$ component contributes nonetheless notably to the $O_2$

changes after the large volcanic eruptions in 1258 and 1452. Furthermore, an anomalously warm phase with relatively low $O_2$ is simulated during the $12^{th}$ and early $13^{th}$ century. AOU anomalies are small over this period and the perturbation in $O_{2,sol}$ dominates during this phase of very weak external forcing (Fig. 1a). Nevertheless, perturbations in AOU are in general much larger than in $O_{2,sol}$ during other PI periods. In contrast, changes in $O_{2,sol}$ contribute for half of the global ocean $O_2$ decrease over the industrial and future period (Fig. 1a). An even larger relative contribution of $O_{2,sol}$ is simulated for the spatially-

averaged thermocline signal (Fig. 1b), while changes in AOU in the thermocline remain small over the industrial period due to compensation of regionally opposed responses, as described below. The $O_2$ variations at the surface are, unsurprisingly, primarily caused by changes in solubility ($O_{2,sol}$). The influence of air-sea $O_2$ disequilibria, included in AOU, is on global average small over the entire duration of the simulation (Fig. 1c).

Following Eq. 1, the natural variability of $O_2$ can be split into the contribution of individual components:

$$Var(O_2) = Var(O_{2,sol}) + Var(-AOU) + 2COV(-AOU, O_{2,sol}) \tag{3}$$

Var() stands for the variance of the variable and equals the square of STD. Both metrics are positive by definition. This implies that the sign of the covariance between $O_{2,sol}$ and -AOU (COV(-AOU, O{2,sol})) indicates whether the contributions from $O_{2,sol}$ and -AOU enhance or partly cancel each other. If COV(-AOU,O{2,sol}) is negative, the resulting Var($O_2$) will be smaller than the sum of Var($O_{2,sol}$) and Var(-AOU).

Table 1 gathers the STD of $O_2$ and its components and the corresponding covariance in the global averaged surface ocean, thermocline and world ocean for the control simulation (CTRL; internal variability) and the forced simulation (LM; natural external and internal variability; 850-1800 CE). COV(-AOU,O{2,sol}) is negative in CTRL in all three water bodies. Hence, $O_{2,sol}$ and –AOU partly compensate each other in CTRL. COV(-AOU,O{2,sol}) is also negative in LM for the surface ocean. This implies that changes in solubility are partly compensated by changes in air-sea $O_2$ disequilibrium. In contrast, COV(-

AOU,O_{2,sol}) is positive in the LM simulation for the global ocean and the spatially-averaged thermocline during the preindustrial period. Hence, changes in $O_{2,sol}$ and –AOU are positively correlated and reinforce each other on average. In





conclusion, the natural forcing leads to a positive correlation of –AOU and $O_{2,sol}$. This statistical analysis is consistent with the discussion on $O_2$ perturbations above: –AOU and $O_{2,sol}$ change hand in hand after large volcanic eruptions in the global mean and in the global thermocline (Fig. 1a, b). The variability in CTRL is not only smaller than in LM, but apparently also different in terms of underlying physical and biogeochemical mechanisms and in their interactions.

Consistent with this, the spatial pattern of the natural variability in $O_2$ (Fig. 3a) can largely be attributed to the natural variability in -AOU (Fig. 7a). $O_{2,sol}$ variations have generally a limited impact on the natural variability of $O_2$ in the thermocline (Fig. 7b). Nevertheless, in the high-variability region in the western North Pacific and the northern North Atlantic, STD($O_{2,sol}$) is of the same order of magnitude as STD(-AOU).

The pattern of STD(-AOU) resembles the pattern of STD for ideal age (Fig. 7a, d). This suggests that a significant fraction of

the variability in -AOU (and $O_2$) is driven by changes in circulation and water mass age. Variability in production of particulate organic matter (POC) in the surface layer (Fig. 7e), indicative of water column remineralisation of organic material, may contribute to the variability in -AOU in the thermocline. For example, in the northern North Atlantic, STD in POC production and -AOU is relatively large, while STD in ideal age is low. On the other hand, the large STD in POC production in parts of the Southern Ocean are not reflected in STD(-AOU). The pattern of STD($O_{2,sol}$) (Fig. 7b) is by definition largely congruent

with the previously discussed pattern of STD(T) (Fig. 3b).

COV(-AOU,O_{2,sol}) shows a large negative amplitude in hot spot regions with large variability in -AOU. These regions are, as discussed in Sect. 3.2 located at the boundaries between major gyres. This suggests that changes in -AOU and $O_{2,sol}$ partly compensate each other in these regions. An exception is the region between the subtropical and subpolar gyres in the North Atlantic, where the two components tend to enhance each other, therefore increase $O_2$ variability.

**3.4.2   Anthropogenic change**

Next, we address the pattern of anthropogenic changes in $O_{2,sol}$ ($\Delta O_{2,sol}$) and in -AOU ($\Delta$(-AOU)) in the thermocline and from PI to the end of the $21^{st}$ century (Fig. 8 and 9). $\Delta O_{2,sol}$ shows a spatially coherent decrease as dictated by the global warming pattern (Fig. 3d). In contrast, $\Delta$(-AOU) shows a strong spatial pattern in the thermocline with positive values in the tropics, Arctic, and subtropical Atlantic and negative values in the mid- and high-latitude Pacific as well as the Southern Ocean

and the subtropical Indian and Pacific Ocean. Regional changes in -AOU largely balance each other, explaining the small change in spatially-averaged -AOU (Fig. 1b). The anthropogenic increase in $O_2$ (Fig. 3c) in parts of the tropical thermocline is attributed to the increase in -AOU, partly offset by the decrease in $O_{2,sol}$, while the anthropogenic $O_2$ decrease in the northern Pacific and Southern Ocean results from a decrease in both -AOU and $O_{2,sol}$.

$\Delta$(-AOU) in the thermocline is mainly driven by changes in ventilation and modulated by changes in remineralisation rates

(Fig. 8c, d). This is similar as for the natural variability in -AOU. The increase in ideal age and in POC production (indicative of an increase in remineralisation rate) explain the decrease in -AOU in the Southern Ocean. By contrast, in the western tropical Pacific, the equatorial Indian and the Atlantic ocean, the combination of a decrease in water mass age and in POC production explains the increase in -AOU over the industrial period and the $21^{st}$ century. In the eastern tropical Pacific and in the North Pacific, the impacts of changes in ventilation are partly mitigated by changes in organic matter remineralisation rate.



Below the thermocline and in the deep ocean, $O_2$ decreases over the industrial period and the $21^{st}$ century as both -AOU and $O_{2,sol}$ decrease (Fig. 9). The decrease in -AOU is again mainly explained by an increase in ideal age. Changes in ideal age in the deep Atlantic, Southern Ocean and Pacific partly exceed 150 years and indicate a general reduction in the deep water mass formation over the industrial period and the $21^{st}$ century in the simulation.

## 3.5 ToE of $O_2$ components

The ToE of oxygen depends on the $O_2$ variability over the last millennium on the one hand and the $O_2$ response to climate change on the other hand. We have shown that in the thermocline, both the variability and the anthropogenic signal of $O_2$ are mainly driven by changes in -AOU modulated by changes in $O_{2,sol}$. The question arises whether the signal of -AOU or of $O_{2,sol}$ emerge earlier than the signal of $O_2$. $O_2$ solubility and thus $O_{2,sol}$ is a function of T, leading to ToE($O_{2,sol}$) very similar to ToE(T) (Fig. 2b) and its difference to ToE($O_2$) (Fig. 2c). Interestingly, the -AOU and $O_2$ signals seem to emerge at the same time in the thermocline in the Pacific and Indian subtropical gyres (Fig. 10b). But the signal of -AOU emerges later in the thermocline in many regions including the subtropical oceans in the Southern Hemisphere, while it emerges earlier in the tropics and in the subtropical gyres of the Atlantic as well as south of Madagascar. This suggests that anthropogenic changes in -AOU might be detectable earlier in large ocean regions. However, the specific results may be model-dependent and it needs to be confirmed by other models or by observations whether the anthropogenic -AOU signal is detectable earlier than the $O_2$ signal in the identified regions. Earlier emergence of -AOU than $O_2$ is generally found in regions with a positive change in -AOU and thus in regions where -AOU and $O_2$,sol partly offset each other. Late emergence in -AOU is found in regions with a small anthropogenic change in -AOU.

## 4   Discussion and Conclusion

We have analysed the variability at interannual timescales and anthropogenic change in marine oxygen ($O_2$) and related physical and biogeochemical variables. We have also determined when the anthropogenic signal leaves the bound of natural variability using the time of emergence (ToE) concept (Hawkins and Sutton, 2012). Results are derived from a simulation performed with the Community Earth System Model (CESM) covering the period 850 to 2100 CE and which is forced with reconstructed volcanic and solar forcing in addition to forcing from land cover changes, greenhouse gases, and other anthropogenic agents (Lehner et al., 2015). This simulation enables to put the anthropogenic changes in the context of the natural forced and internal variability of the last millennium. We find that anthropogenic deoxygenation and warming in the thermocline has today already left the bounds of natural variability. In order to quantify this result, Fig. 11 shows the fraction of the ocean that emergences over the industrial and future period. In 2020, 60 % and 90 % of the thermocline area show anthropogenic deoxygenation and warming respectively. By the end of this century, these values are approaching towards 100 % when greenhouse gas emissions continue unabated with increasing risks for the function and services of marine ecosystem (Pörtner et al., 2014). There are uncertainties in our results, and some are linked to the relatively coarse resolution of the CESM model of order one degree. Larger variability may be found on smaller scales. For example, Long et al. (2016) document that interannual variability



from the Hawaii Ocean Time-Series (HOT) station is about a factor of two larger than the variability at the same location in CESM. Another source of error is structural model uncertainty. Comparison with observations and multi-model studies show weaknesses of the current class of earth system model in simulating the observed $O_2$ distribution and that projections of anthropogenic $O_2$ change are particularly uncertain in low oxygenated waters (Bopp et al., 2013; Cocco et al., 2013).

## 4.1 Natural variability: forced and internal

Both naturally forced and the internal variability contributed to the simulated climate and biogeochemical variations of the last millennium. The internal variability arises from the inherent and partly chaotic variability of the climate system (Baines, 2008; Frölicher et al., 2009; Deser et al., 2012; Resplandy et al., 2015). It is also associated with climate modes such as the El Nino-Southern Oscillation (Bacastow, 1976; Keller et al., 2015), the North Atlantic Oscillation (Keller et al., 2012), the Pacific Decadal Oscillation (Duteil et al., 2018) or the Southern Annular Mode (Hauck et al., 2013). Important natural forced climate variability on interannual to centennial time scales arise from explosive volcanism and from changes in solar irradiance superimposed on long-term trends from orbital variations (Wanner et al., 2008; Jungclaus et al., 2010; Lehner et al., 2015). The comparison between the transient last millennium simulation and the corresponding control simulation shows that both natural forced and internal variability contribute significantly to variations in marine $O_2$ and temperature. While the role of forced variability is particularly large when considering large-scale averages (Table 1, Fig. 4), due to the more important smoothing effect on internal chaotic variability. Large explosive volcanic eruptions cause widespread ocean cooling and positive $O_2$ anomalies. The resulting temperature and $O_2$ perturbations last for decades and centuries in the thermocline and the deep ocean (Fig. 1). This implies that natural forced variations should not be neglected when comparing century-scale anthropogenic climate change to natural climate variability. Yet, earlier studies addressing ToE do not consider natural variability arising from solar and volcanic forcing during the last millennium.

## 4.2 Methodological aspects: Variability estimated from control and industrial period simulations is biased low

ToE is a signal-to-noise problem where a signal of change is compared to the noise of variability. Different assumptions are made in earlier studies (i) to estimate the noise of natural variability (e.g Giorgi and Bi, 2009; Hawkins and Sutton, 2012; Mora et al., 2013; Keller et al., 2014; Rodgers et al., 2015; Carter et al., 2016; Frölicher et al., 2016; Long et al., 2016; Brady et al., 2017; Henson et al., 2017), (ii) the anthropogenic signal ({e.g Rodgers et al., 2015; Carter et al., 2016; Frölicher et al., 2016), (iii) the detection threshold (Rodgers et al., 2015) and (iv) the detection period (Henson et al., 2017). The detection period is typically taken to start around modern times in studies directed to detect the anthropogenic signal by measurements (Henson et al., 2016) or, as in this study, at the beginning of the preindustrial period.

ToE is here taken to be the time when the anthropogenic signal reaches twice the noise of natural variability. In the standard case, the noise is defined as the standard deviation of annual values over the period 850 to 1800 CE. It includes therefore internal and forced variability. We consider annual data as the main focus is on the thermocline where seasonal variability is smaller compared to the surface. The anthropogenic signal is defined in this study as the long-term change relative to year 1800 CE obtained by low-pass filtering the model output with a smoothing spline with a cut-off frequency of forty years (Enting,





1987). We consider this choice as more appropriate than a linear trend (Keller et al., 2014; Henson et al., 2017) because the anthropogenic signal increases in a non-linear way and is highly dependent on the window of time considered Carter et al., 2016. The applied detection threshold of two standard deviations excludes most extreme environmental conditions that are also unusual in the context of the last millennium (Fig. A3), while the application of one standard deviation as a threshold (Rodgers

et al., 2015; Carter et al., 2016; Frölicher et al., 2016) enables a more rapid trend detection.

The way variability is estimated significantly influences estimates of ToE. In this study, ToE for $O_2$ and temperature in the thermocline is estimated using different estimates of variability within a self-consistent setting. Namely, variability is estimated from the forced simulation of the last millennium (850 to 1800 CE) and, alternatively, for a short period (1720 to 1800 CE) and from the control simulation. As expected, using variability from the control and the short period yields in general an earlier

emergence of the anthropogenic signal than when using variability from the last millennium simulation (Fig. 11). These two estimates of noise do not capture the full natural variability of the last millennium. Yet, differences in estimated ToE and noise are often modest on the local scale (Fig. 2, 4 and 11). As last millennium earth system simulations extended towards the future period are still rare, it might be appropriate, though not ideal, to use the results from a control simulation to estimate natural variability.

In addition, we have tested the use of the "residual" variability of a time series to estimate noise. The long-term trend in the industrial period (1800 - 2005 CE) data is removed by a low pass filter; the standard deviation of the remaining annual anomalies provides then an estimate of noise. This yields a much smaller variability than estimated from the last millennium output and a much earlier ToE (Fig. 6 and 11). Applying a cut-off period of 80 and 100 years instead of 40 years for the smoothing spline leads to higher variability but still underestimates the last millennium variability (not shown). The residual

variability is readily estimated from existing and forthcoming measurement time series (e.g. Hawaii Ocean Time-series, Ocean Station Papa, Bermuda Atlantic Time-series Study), while temporally resolved ocean biogeochemical data are missing for the last millennium. Another published approach is to estimate linear trends and their uncertainty from an ensemble of model simulations (Rodgers et al., 2015; Frölicher et al., 2016; McKinley et al., 2016). This approach is insightful. However, there is only one realisation of the real climate system evolution and sufficiently long time series to estimate the uncertainty in

long-term trends from observations are largely missing.

We have shown that the absolute ToE values are noise- and signal-definition dependent. The comparison across the published ToE analyses demonstrated as well a model dependency. However, the relative ToE (ToE(T)-ToE($O_2$)) shows similar patterns and values across all the methods described and tested in this study (not shown). We conclude that the use of ToE of one variable in comparison to the ToE of another variable may bring more robust insights on the ToE analysis.

### 4.3 Anthropogenic deoxygenation is earlier detectable than anthropogenic warming in some regions

We find ToE for $O_2$ to be early in the mid- and high latitude thermocline and late in the tropical ocean and subtropical Atlantic, in agreement with earlier work (Long et al., 2016; Henson et al., 2017). ToE for temperature has not been analysed in the thermocline in earlier studies. In CESM, ToE for temperature is early in eastern boundary systems, the subtropical Atlantic and in large parts of the Southern Ocean, but late in many subtropical regions and in large parts of the North Pacific. A large





$O_2$ decline is simulated by CESM, in agreement with other models (Bopp et al., 2013; Cocco et al., 2013), in the North Pacific thermocline, where changes in ventilation are particularly large. As an interesting consequence, the anthropogenic $O_2$ signal is earlier detectable than the anthropogenic temperature signal in large parts of the northern North Pacific. Keller et al. (2015) show that a potential weakening of the ENSO variability is earlier verifiable and more widespread for carbon cycle tracers than

for temperature and Séférian et al. (2014) highlight the multi-year predictability of tropical productivity. This corroborates earlier suggestions by Joos et al. (2003) that measurements of $O_2$, or more general multi-tracer observations, are instrumental to better detect or predict anthropogenic change.

### 4.4 Anthropogenic trend for AOU may be earlier detectable than for $O_2$ in some regions

We have presented results in terms of variability and anthropogenic change not only for $O_2$ and temperature, but also for

variables indicating the role of physical and biological mechanisms. These are apparent oxygen utilisation (AOU), indicative of changes in the marine biological cycle, the solubility component of $O_2$ ($O_2$,sol), indicative of thermally (and salinity) driven changes in $O_2$, ideal age, indicative of circulation changes and water mass ventilation, and production of particulate organic carbon (POC), indicative of biological export and the amount of organic matter remineralised in the water column.

Most of the last millennium variability in $O_2$ is explained by variations in AOU and by changes in water mass ventilation,

with a smaller role for solubility changes and changes in biological productivity. Variability in $O_2$, AOU and ideal age is particularly large in boundary regions of the major gyres. In such regions, variations in AOU and $O_{2,sol}$ partly compensate each other as revealed by the covariance between the two variables. Turning to anthropogenic change, changes in AOU and $O_{2,sol}$ partly cancel each other in the tropical thermocline and in the subtropical Atlantic in accordance with Bopp et al. (2017). This results in an earlier ToE of AOU than $O_2$ in parts of the tropical ocean. This suggests that anthropogenic change in AOU

may be earlier detectable by measurements than in $O_2$ in specific ocean regions.

### 4.5 Large deoxygenation in the deep ocean ahead

$O_2$ changes will likely continue beyond 2100 CE. We find a strong link between changes in $O_2$, AOU and ideal age, with a shift to older water mass ages accompanied by a shift to lower $O_2$. By 2100, ideal age in the near bottom waters of the Southern Ocean and the deep Pacific has increased by up to 240 years and $O_2$ decreased by around 16 to 20 $\mathrm{mmol\,m^{-3}}$ relative to

preindustrial. These age and $O_2$ anomalies are likely to spread further into the deep ocean. A long-term reduction in deep ocean ventilation and $O_2$ under anthropogenic forcing is consistent with results from Earth System Models of Intermediate Complexity (Schmittner et al., 2008; Battaglia and Joos, 2018). For example, Battaglia and Joos (2018) find a large, transient decline in deep ocean $O_2$ and in the global $O_2$ inventory by as much as 40 % in scenarios where radiative forcing is stabilised in 2300 CE. In their simulations, deoxygenation peaks about a thousand years after forcing stabilisation and new steady-state conditions

are established only after 8000 CE. The CESM results also support the notion of a long-term deep ocean deoxygenation. In conclusion, we find natural radiative forcing arising from explosive volcanism and solar irradiance changes to contribute notably and in addition to internal climate variability to the overall natural variability in marine $O_2$ and temperature. We simulate





large and widespread ocean deoxygenation under anthropogenic forcing and suggest that large parts of the thermocline already experience environmental conditions that are outside the range of natural variability of the last millennium.





## 5 Figures





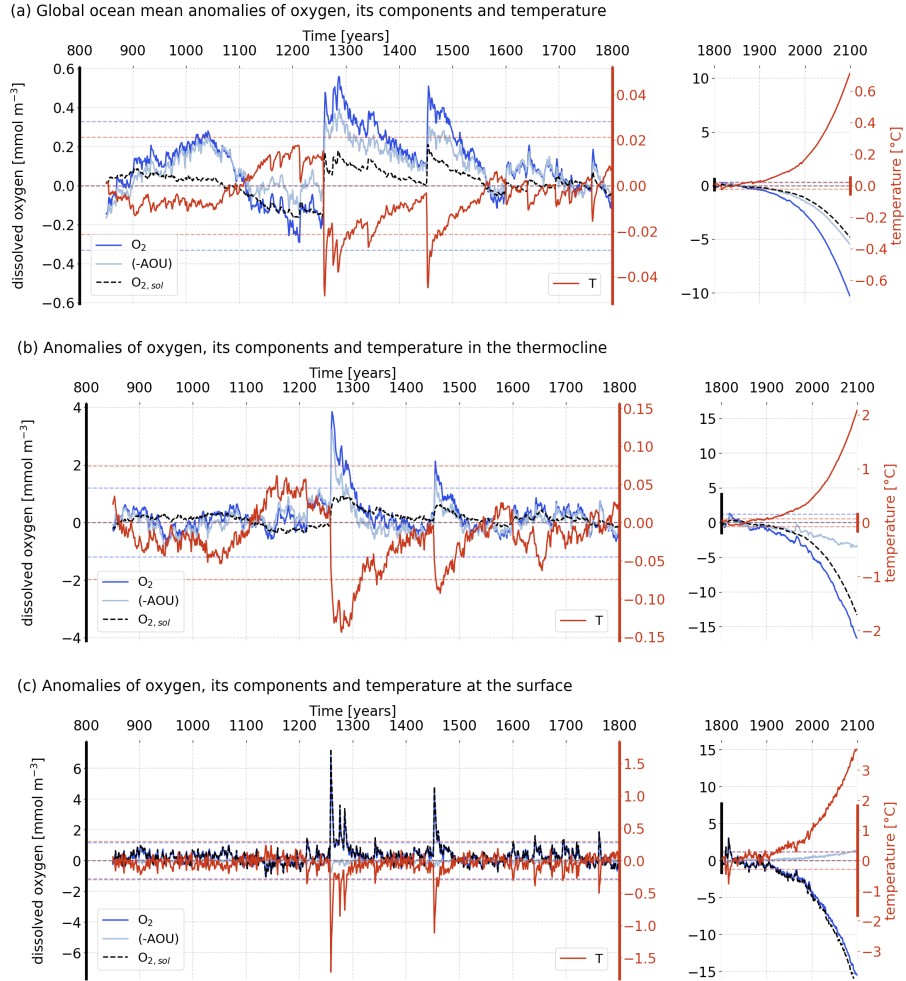

**Figure 1.** Temporal evolution of simulated anomalies in (a) global mean $O_2$ (dark blue), -AOU (light blue), the $O_2$ solubility component (dashed black), and T (red) over the last millennium (left) and the historical and future period (right). The horizontal dashed lines stand for the two standard deviations envelops for O2 (blue) and T (red) computed over the period 850-1800 CE. (b) and (c) same as (a) but for the thermocline (200-600 m) and the surface, respectively. The anomalies are relative to the pre-industrial reference period (1720-1800 CE).





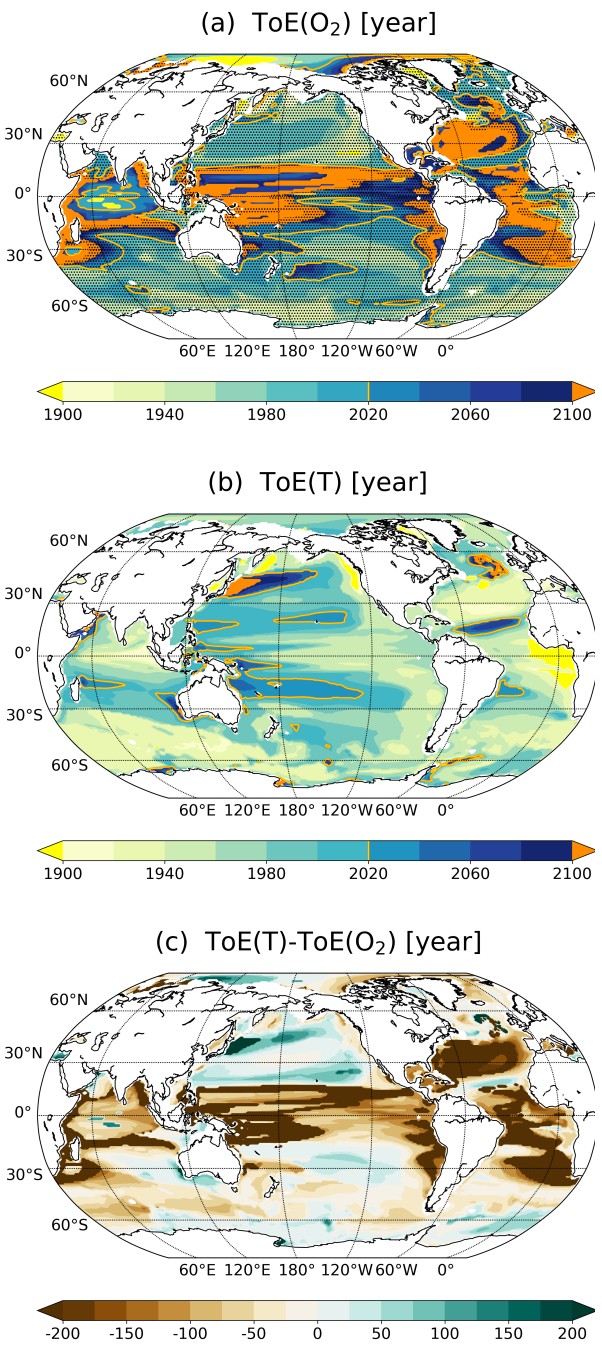

**Figure 2.** Time of Emergence (ToE) of (a) $O_2$, (b) T and (c) their difference in the thermocline (200-600 m). The dashed areas in (a) indicate where $O_2$ decreases under RCP8.5. Regions where the anthropogenic signal has not emerged by the end of the $21^{st}$ century are indicated by orange.



**Figure 3.** Standard deviations (STD) for (a) $O_2$ and (b) T as computed for the period 850 to 1800 CE for the thermocline (200-600 m). Anthropogenic changes ((2070-2099) minus (1720-1800)) in (c) $O_2$ and (d) T in the thermocline. $O_2$ and T are annually and vertically averaged and STD and changes are computed from these averaged values. The magenta arrows in (c) and (d) indicate the section shown in Fig. 9 (Atlantic: 25° W; Southern Ocean: 60° S; Pacific: 150° W).



**Figure 4.** Ratio of the standard deviations (STD) from the forced simulation (0850-1800 CE) versus those from the control simulation (CTRL) for $O_2$ (left column) and T (right column) for (a, b) the global ocean, (c, d) the thermocline (200-600 m) and (e, f) the surface. The magenta contours highlight the ratio equal to 1, i.e., where STD are equal in the forced and control simulation. STD are computed from annually (all panels) and vertically (panels a-d) averaged values.





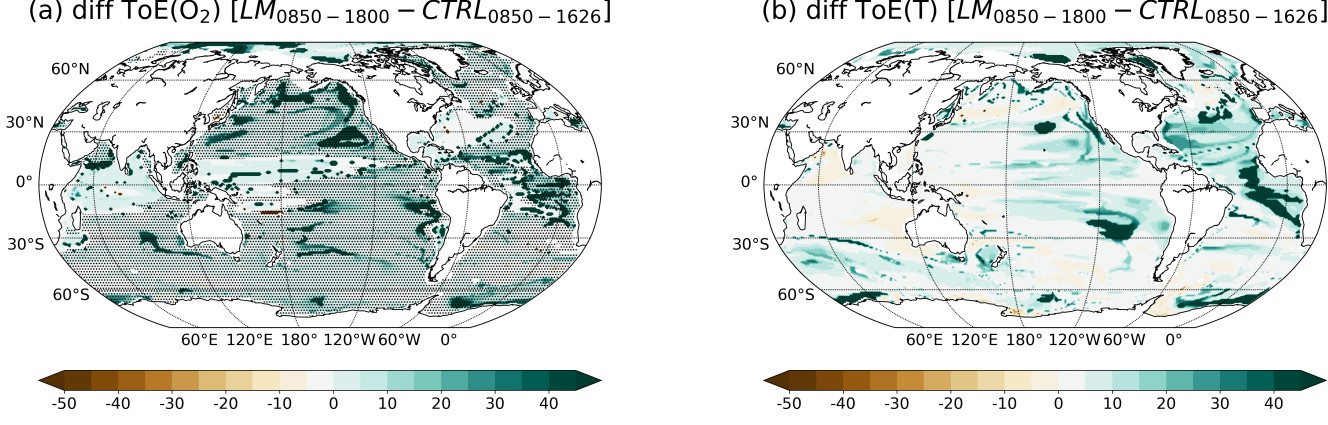

**Figure 5.** Difference of ToE for (a) $O_2$ and (b) T using the STD of the forced simulation (0850-1800 CE) minus the ToE using the STD of the control simulation. The dashed areas in (a) indicate where $O_2$ decreases under RCP8.5.





**Figure 6.** Ratio of the standard deviations (STD) computed for the period 850-1800 CE versus (a, b) those for 1720-1800 CE from the forced simulation and versus (e, f) the residual variability for $O_2$ (left panel) and T (right panel). Difference of ToE using the STD of the forced simulation during the last millennium (0850-1800 CE) minus (c, d) the ToE using the pre-industrial period (1720-1800 CE) and (g, h) the ToE using the residual variability for $O_2$ (left panel) and T (right panel). The magenta contours highlight the ratio equal to 1, i.e., where STD are equal. STD are computed from annually and vertically averaged values between 200 and 600 m. The dashed areas in (c, g) correspond to regions where oxygen decreases under RCP8.5.



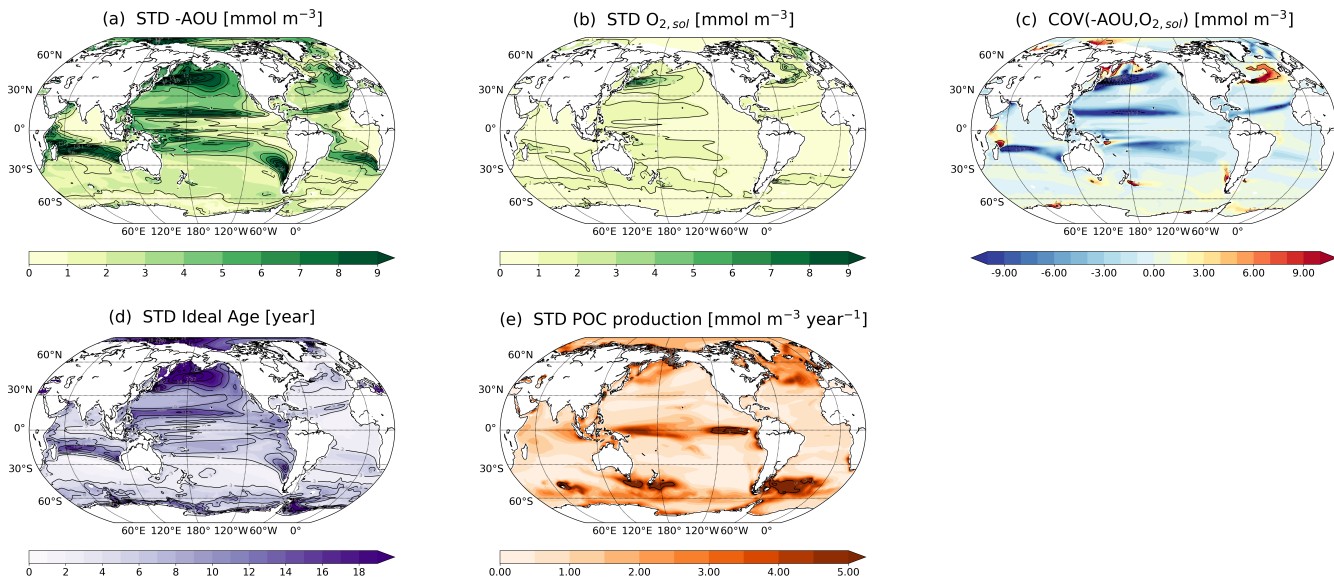

**Figure 7.** Standard deviation of annually and vertically-averaged (200-600 m) (a) -AOU, (b) $O_{2,sol}$, (d) ideal age, and (e) Particulate Organic Carbon production (POC production) during the last millennium (850-1800 CE) and (c) the corresponding covariance between -AOU and $O_{2,sol}$.





**Figure 8.** Anthropogenic changes ((2070-2099 CE) minus (1720-1800 CE)) in (a) -AOU, (b) $O_{2,sol}$, (c) ideal age and (d) Particulate Organic Carbon production (POC production). The results are averaged between 200 and 600 m, except for POC production which is averaged between the surface and 200 m. The magenta arrows indicate the section path used in Fig. 9 (Atlantic: 25° W; Southern Ocean: 60° S; Pacific: 150° W).





(a) ΔO$_2$ [mmol m$^{-3}$]

(b) ΔT [°C]

(c) Δ(-AOU) [mmol m$^{-3}$]

(d) ΔO$_{2,sol}$ [mmol m$^{-3}$]

(e) ΔIdeal age [year]

(f) ΔPOC production [mmol m$^{-3}$ year$^{-1}$]

**Figure 9.** Anthropogenic changes ((2070-2099 CE) minus (1720-1800 CE)) in (a) O$_2$, (b) T, (c) -AOU, (d) O$_{2,sol}$, (e) ideal age and (f) POC production. The sections are taken along 25° W in the Atlantic, 60° S in the Southern Ocean and 150° W in the Pacific (Fig. 3b and d and in Fig. 8, magenta arrows).





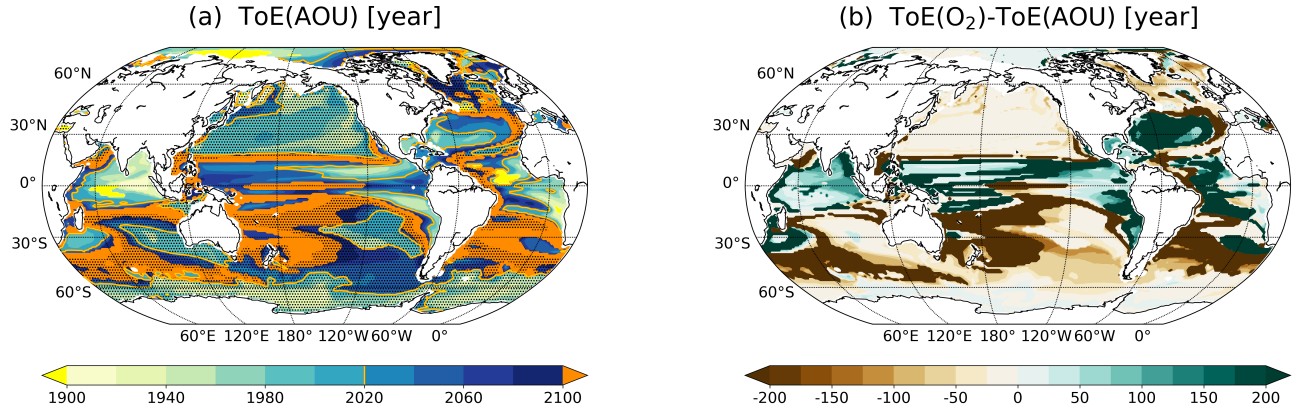

**Figure 10.** Time of Emergence (ToE) of (a) -AOU, (b) $O_2, sol$ in the thermocline (200-600 m). The dashed areas in (a) indicate where -AOU decreases under RCP8.5. Regions where the anthropogenic signal has not emerged by the end of the $21^{st}$ century are indicated by orange for AOU. (b) Difference of the ToE for $O_2$ and the ToE of -AOU in the thermocline (200-600 m).





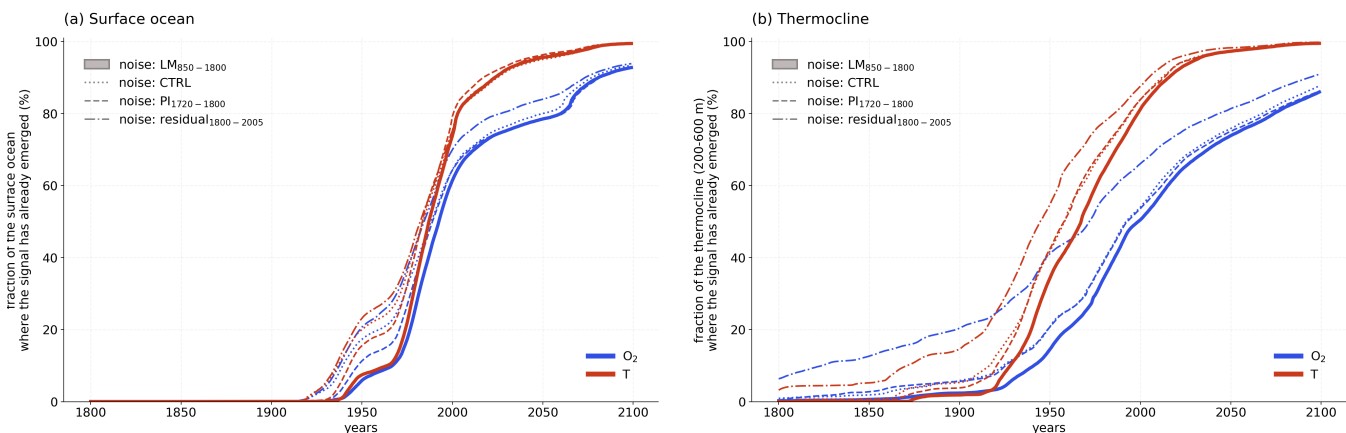

**Figure 11.** Fraction of the (a) surface ocean and (b) thermocline (200-600 m) where the signal has already emerged in oxygen (blue) and temperature (red) for different definitions of the background noise: (i) naturally forced variability during the last millennium (850-1800 CE, LM; solid), (ii) internal variability (CTRL; dotted), (iii) naturally forced variability during the pre-industrial period (1720-1800 CE, PI; dashed), (iv) residual variability using a low pass filter (cut off period of 40years, residual; dashed dotted).





|  |  | Temperature | O$_2$ | -AOU | O$_{2,sol}$ | COV(-AOU,O$_{2,sol}$) |
|---|---|---|---|---|---|---|
|  |  | [°C] | [mmol m$^{-3}$] | [mmol m$^{-3}$] | [mmol m$^{-3}$] |  |
| Global ocean | CTRL | 0.3 10$^{-2}$ | 3.6 10$^{-2}$ | 4.9 10$^{-2}$ | 3.4 10$^{-2}$ | -1.13 10$^{-3}$ |
|  | LM | 1.1 10$^{-2}$ | 16.5 10$^{-2}$ | 10.7 10$^{-2}$ | 7.2 10$^{-2}$ | 5.30 10$^{-3}$ |
| Thermocline (200-600 m) | CTRL | 1.3 10$^{-2}$ | 36.9 10$^{-2}$ | 36.4 10$^{-2}$ | 8.9 10$^{-2}$ | -2.13 10$^{-3}$ |
|  | LM | 3.7 10$^{-2}$ | 59.8 10$^{-2}$ | 48.2 10$^{-2}$ | 24.4 10$^{-2}$ | 32.9 10$^{-3}$ |
| Surface | CTRL | 6.4 10$^{-2}$ | 27.4 10$^{-2}$ | 8.4 10$^{-2}$ | 28.2 10$^{-2}$ | -5.75 10$^{-3}$ |
|  | LM | 14.6 10$^{-2}$ | 58.8 10$^{-2}$ | 10.5 10$^{-2}$ | 63.5 10$^{-2}$ | -34.3 10$^{-3}$ |

Standard deviation column spans Temperature, O$_2$, -AOU, O$_{2,sol}$.

**Table 1.** Overview of the standard deviations of T, O$_2$, O$_{2,sol}$ and -AOU, and the corresponding covariance between -AOU and O$_{2,sol}$ for the global mean ocean, the thermocline (200-600 m) and the surface ocean for the control simulation (CTRL) and the forced simulation (850-1800; LM). STD and COV are computed from annually and spatially averaged data.

## Appendix A: Supplementary material

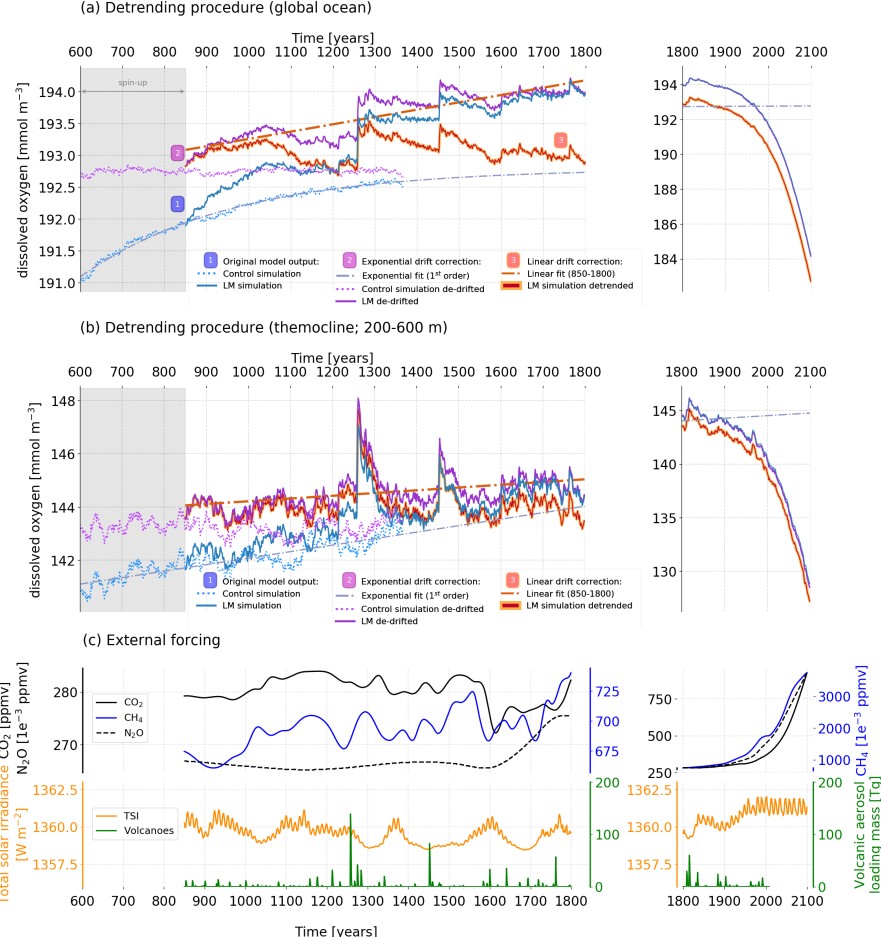

**Figure A1.** Illustration of the two-step procedure to remove model drift and millennial scale trends at each grid point shown in (a) the global mean ocean and (b) the thermocline (200-600 m). The original, annually-averaged outputs from the control (CTRL) and the forced simulations are shown by dashed and solid blue lines, respectively [1]. The results of the CTRL are fitted by an exponential function with one time scale and extrapolated towards equilibrium (dashed line). The exponential curve is subtracted from and the equilibrium value added to the original outputs to obtain the purple curves [2] (dotted : CTRL, solid : forced simulation). The remaining multi-millennial trend in the forced simulation is removed using a linear fit (850-1800 CE), leading to the red curve [3].

(c) Time series of the main external forcings applied to the forced simulation: total solar irradiance [W m$^{-2}$] at the top of the atmosphere in orange, stratospheric volcanic aerosol load [Tg] in green and the greenhouse gases $CO_2$ (solid black line), $N_2O$ (dashed black line) and $CH_4$ (blue solid line) [ppmv].







**Figure A2.** (a) Oxygen concentration from the data-based World Ocean Atlas (Garcia et al., 2013), (b) temperature observed from observation-based (Locarnini et al., 2013), the simulated (c) oxygen and (d) temperature by the model CESM during the period 1986-2005. The maps show averages between 200 and 600 m.



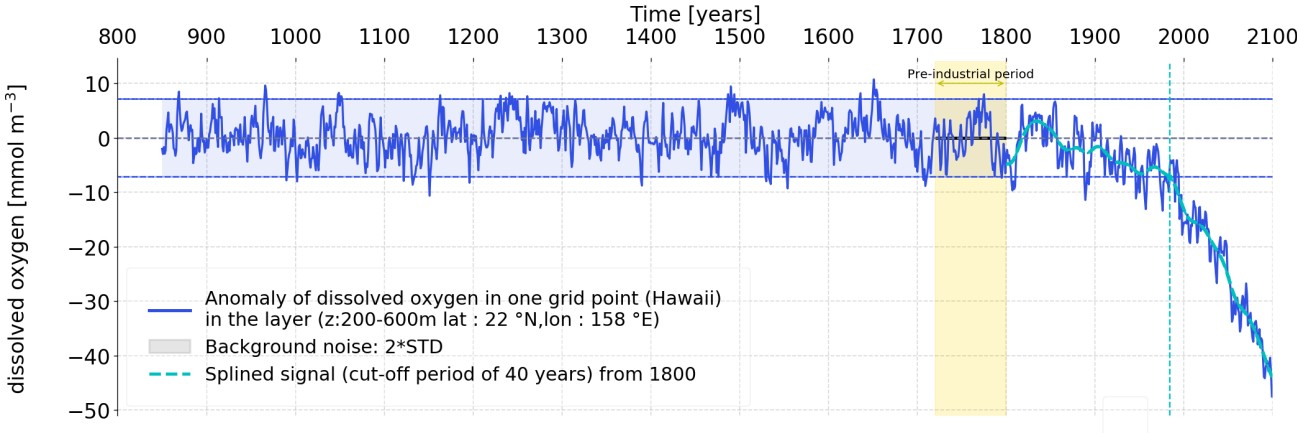

**Figure A3.** Illustration of the Time of Emergence (ToE) method. The example is for the thermocline (200-600 m) and grid cells located at 158° W and 22° N near Hawaii. The standard deviation (STD) of the detrended, annually and vertically (200-600 m) averaged data (blue) for the period 850-1800 CE is used to define the "noise" or bounds of natural variability which is set to ±2STD (blue area). The annually and vertically averaged data are fitted with a spline using a cut-off period of 40 years (light blue). ToE is the point in time when the spline crosses and leaves the bounds of natural variability; here, ToE is 1984 and indicated by the vertical dashed line. All data are anomalies relative to the preindustrial period 1720-1800 CE (yellow).





*Acknowledgements.* This study is funded by the Oeschger Centre for Climate Change Research and the Swiss National Science Foundation (#200020_172476). We thank C. Raible and F. Lehner for providing CESM output and T. Frölicher and O. Aumont for discussion.





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
