# Peer review of "Assessment of time of emergence of anthropogenic deoxygenation and warming: insights from a CESM simulation from 850 to 2100 CE"

_Biogeosciences, 2018_

## Referee Comment (RC1) · Anonymous Referee #1 · 27 Feb 2019

General comments

The manuscript addresses the important issue of climate signal emergence in ocean oxygen concentrations using the Community Earth System Model (CESM). The authors employ the widely used Time of Emergence (ToE) signal-to-noise metric. There is a new focus in the analysis on the impacts of natural external forcings such as volcanism over the last millennium, and projections are analysed over the 21st century (for dissolved oxygen and temperature fields). The work is accomplished, certainly relevant to the scope of BG, and some interesting conclusions are reached regarding spatial patterns in Time of Emergence (ToE). The methods are appropriate and well

**BGD**

justified, including a careful discussion of the de-trending (de-drifting) procedure. It's worth noting that this work is not entirely novel, and as the authors point out many other similar ToE studies have been done on CMIP5-class models, but as above I'm confident that the focus on natural external forcing and the last millennium is sufficiently distinct to justify publication. However I have two general issues which require the authors attention in revision along with specific / technical comments listed below.

1. Primarily the manuscript requires major revision in terms of language and readability. The paper would be significantly enhanced by a reduced wordcount and tightening of the text, along with a close check for typos and grammar. There is need for improvements to sentence structure throughout. I have flagged some specific issues in the Technical Comments but this is not comprehensive and will need to be re-reviewed with this in mind. In addition, I suggest that the sections of the Results which test ToE methodological assumptions such as noise estimates (e.g. Sect. 3.3.2 – 3.3.3) should be (at least) condensed or potentially moved to the Supplement. The Discussion also includes lots of sub-sections and is not very readable in its present form.

Some of this is certainly a matter of opinion, but it would be great to see a more streamlined manuscript which focuses on the central ToE results and the separation of O2 changes into AOU, solubility components.

Also there are lots of maps in the Results which do not necessarily add much to the paper (e.g. Fig 4. panel) – are they all necessary? e.g. do we need surface, thermocline and full depth averaged maps? (see below).

2. The authors spend a lot of time talking about uncertainties in noise estimates and methodological approach, however only briefly mention (Sect. 4) the potentially major influence of model structural error in their analysis. Specifically, the study does not validate the CESM simulated pattern of historical oxygen change using observations (only the mean state in Fig A2) or compare the magnitude of simulated internal variability to e.g. long-term oxygen time series data. It is likely that the widely reported

[Figure]

(e.g. Stramma et al., 2012) lack of model-data agreement in reproducing observed low-latitude deoxygenation plays a major role in the authors reported (low) ToE in the tropics (e.g. Fig 2a). This limitation is true of all ToE studies and somewhat unavoidable, however it needs to be explored more as a major source of uncertainty in the study. To this end I suggest the authors include a comparison of CESM simulated historical deoxygenation to observations (e.g. vs the global Schmidtko et al. [2017] dataset) since this influences considerably the interpretation of ToE estimates derived for the 21st century.

The authors also note in Sect. 4 that observed variability in oxygen (at HOTS) may be a factor of two larger than simulated by CESM. Along with the forced trend mentioned above if the noise is underestimated this will considerably impact upon CESM derived ToE estimates. The authors also need to more thoroughly address this source of uncertainty in the manuscript.

Specific comments

Line 9 – 10: "natural variability [. . .] are systematically larger than internal variability". This needs to be clearer. I think you mean control estimates are not good enough as they don't include natural external forcings like volcanism, not that model simulated variability is smaller than observed? (or both?)

Page 1 Line 12: do you mean "anoxic" or suboxic?

Introduction Page 1 line 18. Do you need to reference all these studies to introduce the well-known concept of ventilation age? And then the next statement that warming leads to solubility driven deoxygenation is not referenced.

Page 2 Line 5: Long et al. do not use optimal fingerprinting in their assessment. Please distinguish between studies which use optimal fingerprinting and ToE (these approaches are substantially different since optimal detection studies include an observed change and ToE are primarily model based)

Page 2 Line 6: For completeness there is another ToE ocean biogeochemistry study by Christian (2014) PLOS ONE

Page 3 Line 10. As the authors note, multiple studies have done ToE on biogeochemistry for CMIP5-class models. It is necessary in the Introduction to highlight what this study does differently and why it is important. I'm confident this can be done e.g. focus on natural external forcings, millennial scale simulations etc.

Page 3 line 32: here and elsewhere I think the use of the terms like "natural variations in external forcing factors" is confusing. Please use concise, clear terminology e.g. "natural external forcings"

Page 8. Section 2.2.3: More justification for the chosen standard deviation noise thresholds is required along with reference to the associated statistical confidence levels required.

Figure 1. The different y-axes between left (last millennium) and right (future forcings) and upper (global) and lower (surface) are understandable but confusing given the amount of detail (AOU vs O2sol) in the panel. This Figure needs to be reworked for clarity to focus on key results. E.g. is surface and full depth averaged oxygen concentration important? Or just the thermocline?

Page 9 – 10 Sect. 3.2. and page 10 – 11 Sect 3.2. These sections are overlong and should be more concise to focus on the key messages.

Figure 4. There is not much added information in this panel – could the Figure be condensed ?

Figure 5, Sect 3.3.2 and Figure 6, Section 3.3.3: Suggest to (at least) condense these predominantly methodological Sections.

Page 13 Sect. 3.4.1 Lines 8 to 18. I suggest to move this extra analysis of Fig 1 back to Sect 3.1 or simplify Figure 1 and add another Figure here to look at AOU/solubility contributions.

Page 16 Line 1 - 4. See overall comments on model uncertainty regarding noise and forced response.

Section 4. The Discussion should be re-written to be more concise and focused.

Technical comments

Abstract Line 1. "aggravate" is a little unclear here suggest to reword

Introduction and throughout: many references are used to support each point. Please be more selective – 13 references for one statement is quite a lot.

Introduction Page 1 line 20. Suggest to cite Capotondi et al. 2012 on stratification

Introduction Page 3 line 22. "by appropriately adding [..]" this is unclear please rework sentence

Introduction Page 3 line line 25 "are considered not at all" rephrase please

Introduction Page 3 line 33 "these externally forced variability" rephrase please

Introduction Page 3 line 34 repetition of "last millennium climate simulations" rephrase please

Introduction Page 4 line 12 remove "the" before 'anthropogenic"

Page 4 line 25: please rephrase (also replace with "rely")

Page 5 line 3. Replace "stands for"

Page 5 line 19. Of carbon?

Page 6 line 5: Should supplemental figures be named in the order they are introduced? (applies to later sections)

Page 10. Replace "outweight" with "exceed"

Page 11 Line 26 "larger"

Page 13 – 14. Formatting issues with subscripts

Page 15 Line 25 missing word after "enables"

Page 17 Line 2 . Missing parentheses

---

## Referee Comment (RC2) · Anonymous Referee #2 · 3 Mar 2019

The authors has presented an interesting study, with a number of valuable analyses and interpretations regarding the Time of Emergence for oxygen and temperature in the ocean. Although the material presented in the manuscript should be of value to the broader research community, it would benefit greatly from revisions to improve clarity, and also to become better anchored in a discussion of existing scientific literature. Suggestions for improving the scientific clarity and impact are detailed below.

MAIN COMMENTS

The Introduction is not sufficiently focused and was a bit all over the map (encyclopedias on multiple topics), and should be streamlined. It would be of great value to state

clearly why trend detection is priority when interpreting observations, and to relate this to the realm of uncertainty in climate change projections. There are several sources of uncertainty and/or ambiguity in trend detection relating to the "noise" component of trend detection. One issue (that emphasized in the manuscript) is the distinction between natural and internal variability, with there being a need to understand and quantify this distinction. This analysis is great, and warrants emphasis. But a second issue is related to the way in which noise is calculated. In most studies that also emphasize initial condition large ensemble methods, the method of Deser et al. (2014) is used to estimate noise, with this typically involving linear trends calculated over decades rather than STD of annual means to calculate ToE. As the amplitude of inter annual and decadal variability is typically expected to be distinct, at face value it is not obvious how to connect the estimates give here with more general research using large ensemble simulations. It's very important to emphasize this point, while it seems also OK to point out that there is nothing inherently flawed or wrong with the method proposed in the manuscript, it is just somewhat different, and more similar to methods that have been applied cross multiple models in inter comparison studies.

As a related point, I believe it would be valuable to communicate the implications of this study to the broader community, given the discrepancies noted above. Is there a way to bridge the different methodologies with large ensemble methods, perhaps by sampling (randomly?) decadal trends from a 1000+ year Last Millennium simulation? That would still be different from what is done with large ensemble runs. Do the authors recommend at all major modeling centers that are embarking on large ensemble simulations also include Last Millennium simulations? Or for CMIP6 protocols (where the historical period goes through 2014) can potential biases be estimated by comparing full historical ensemble runs with greenhouse CO2-only (for example, for estimating internal variability) runs over 1850-2014?

MINOR COMMENTS

(1) References Near the top of Pg. 3 the authors refer to Hawkins and Sutton (2012),

where the methods considered for calculating Time of Emergence are not the same as those typically used with large ensemble methods (Deser et al., 2014). This should be clarified with regard to the comments above. Also, are the authors sure that the Long et al. (2016) paper used the same method to calculate noise as that of Hawkins and Sutton?

(2) Model configuration Is the CAM4 model considered here the same as the atmospheric model component used by Kay et al. (2015)? More generally, how do the model components differ, and if so, how might this itself impact variability?

(3) ToE, Pg. 9, line 8 and ensuing paragraph The section header "ToE" needs to be expanded into "Time of Emergence (ToE)", or something similarly appropriate. It should also be stated explicitly in this paragraph which year is used as a "reference" for the ToE calculations (ToE relative to what year?)

(4) Fig. 2: Pg. 9, describing text The patterns and timescales should be compared with existing published literature for oxygen and temperature, with any caveats about the methods used to calculated noise.

(5) LM, pg 11, noie 25: The authors need to spell out LM as Last Millennium here.

(6) pg 11, lines 30-34 It would improve the clarity of the presentation if a bit more detail were provided here. What are the percent differences, and over what regions?

(7) Section 4.1 on pg 16 The references described here are not appropriate for linking biogeochemistry and climate modes, with the exception of the Bacastow reference. For the case of ENSO it would be appropriate to reference the study of Winguth et a l. (1990s?). I believe for the SAM there were the studies in 2006-2007 of LeQuere, Lenton, and Lovenduski, and for the PDO you might consider the study of McKinley (2006).

---

## Author Comment (AC1) · 5 Apr 2019

We thank both reviewer for assessing this manuscript and for their time and effort. The useful comments are much appreciated and helped to improve the presentation of our results. Please find attached to this reply a revised manuscript where planned text changes are highlighted.

Please also note the supplement to this comment:
https://www.biogeosciences-discuss.net/bg-2018-523/bg-2018-523-AC1-supplement.pdf

---

## Author Response (AR1)

**Reply to review comments**

We thank both reviewers for assessing this manuscript and for their time and effort. The useful comments are much appreciated and helped to improve the presentation of our results. Conclusions remain unchanged.

The original review comments are given below in black, our reply in blue, and quotes from the revised manuscript in gray. Please find attached to this reply a revised manuscript where text changes are highlighted.

**1 Reviewer 1**

General comments
The manuscript addresses the important issue of climate signal emergence in ocean oxygen concentrations using the Community Earth System Model (CESM). The authors employ the widely used Time of Emergence (ToE) signal-to-noise
5 metric. There is a new focus in the analysis on the impacts of natural external forcings such as volcanism over the last millennium, and projections are analysed over the 21st century (for dissolved oxygen and temperature fields). The work is accomplished, certainly relevant to the scope of BG, and some interesting conclusions are reached regarding spatial patterns in Time of Emergence
10 (ToE). The methods are appropriate and well justified, including a careful discussion of the de-trending (de-drifting) procedure. It's worth noting that this work is not entirely novel, and as the authors point out many other similar ToE studies have been done on CMIP5-class models, but as above I'm confident that the focus on natural external forcing and the last millennium is sufficiently
15 distinct to justify publication. However I have two general issues which require the authors attention in revision along with specific / technical comments listed below.
Thank you for your general support and your constructive comments.

20 1. Primarily the manuscript requires major revision in terms of language and readability. The paper would be significantly enhanced by a reduced word-count and tightening of the text, along with a close check for typos and grammar. There is need for improvements to sentence structure throughout. I have flagged some specific issues in the Technical Comments but this is not comprehensive
25 and will need to be re-reviewed with this in mind.
We have carefully checked the manuscript for grammar and typos, with the help

of native English speaking colleagues.

In addition, I suggest that the sections of the Results which test ToE methodological assumptions such as noise estimates (e.g. Sect. 3.3.2 – 3.3.3) should be (at least) condensed or potentially moved to the Supplement. The Discussion also includes lots of sub-sections and is not very readable in its present form. Some of this is certainly a matter of opinion, but it would be great to see a more streamlined manuscript which focuses on the central ToE results and the separation of O2 changes into AOU, solubility components.

We have substantially shortened section 3.3 and reduced the number of corresponding figures. Please see the attached manuscript for details. Nevertheless, we consider that the comparison between the different estimates of variability and their implications for estimating ToE is an essential and novel element. Studies that compare internal and naturally forced variability are rare and missing for $O_2$ and T. We have also largely rewritten the discussion, and removed the subsections.

Also there are lots of maps in the Results which do not necessarily add much to the paper (e.g. Fig 4. panel) – are they all necessary? e.g. do we need surface, thermocline and full depth averaged maps? (see below).

We followed the suggestion:
-moved Fig. 6a, 6b, 6c, and 6d to the appendix
-removed the full depth averaged analysis (Fig. 4a and 4b).

2. The authors spend a lot of time talking about uncertainties in noise estimates and methodological approach, however only briefly mention (Sect. 4) the potentially major influence of model structural error in their analysis. Specifically, the study does not validate the CESM simulated pattern of historical oxygen change using observations (only the mean state in Fig A2) or compare the magnitude of simulated internal variability to e.g. long-term oxygen time series data. It is likely that the widely reported (e.g. Stramma et al., 2012) lack of model-data agreement in reproducing observed low-latitude deoxygenation plays a major role in the authors reported (low) ToE in the tropics (e.g. Fig 2a). This limitation is true of all ToE studies and somewhat unavoidable, however it needs to be explored more as a major source of uncertainty in the study. To this end I suggest the authors include a comparison of CESM simulated historical deoxygenation to observations (e.g. vs the global Schmidtko et al. [2017] dataset) since this influences considerably the interpretation of ToE estimates derived for the 21st century.

The authors also note in Sect.4 that observed variability in oxygen (at HOTS) may be a factor of two larger than simulated by CESM. Along with the forced trend mentioned above if the noise is underestimated this will considerably impact upon CESM derived ToE estimates. The authors also need to more thoroughly address this source of uncertainty in the manuscript.

The simulated historical $O_2$ concentrations are now compared to the data set

from Schmidtko et al. (2017) (Fig. A3). Moreover, we refer to Long et al. (2016) who compared CESM large ensemble results with the time series from BATS, HOT, OSP stations. Though their analysis is done on isopycnal surfaces, the conclusion concerning the amplitude of the natural variability applies to our study. The corresponding text in section 2.1.3 reads:

The evaluation of the modelled variability remains difficult as observational data are sparse. Furthermore, there is an inherent mismatch in the spatial scale between local measurements and the model resolution, which is of the order of 100 km. The modelled data are thus spatially averaged as compared to the observations and their variability does not explicitly take into account mesoscale eddies and other small scale processes (Long et al., 2016). Figure A3 compares the reconstructed variability and trends in $O_2$ over recent decades at 300 m depths and for different basins (Schmidtko et al., 2017) with the model results. CESM simulated historical $O_2$ concentrations show multi-decadal variability, although with a much smaller amplitude compared to the observations. Long et al. (2016) have compared historical time series from the Hawaii Ocean Time-Series (HOT) station, the Ocean Station Papa (OSP) and the Bermuda Ocean Time Series (BATS) to the CESM large ensemble. They show that modelled variability in annually-averaged $O_2$ is substantially smaller than observed. Taken at face value, beyond the limitations described above, these comparisons suggest that variability in CESM may be biased low. This would tend to bias ToE towards early emergence.

We also slightly modified and expanded the paragraph where uncertainties are discussed (Sect. 4, p16):

However, there are uncertainties in our results, and some are linked to the relatively coarse resolution of the CESM model of order one degree. Larger variability may be found on smaller scales. For example, Long et al. (2016) document that interannual variability from the Hawaii Ocean Time-Series (HOT) station is about a factor of two larger than the variability at the same location in CESM. Another source of error is structural model uncertainty. Comparison with observations (Sect. 2.1.3) and multi-model studies show weaknesses of the current class of earth system models in simulating the observed $O_2$ distribution and variability. Projections of anthropogenic $O_2$ change are particularly uncertain in low oxygenated waters (Bopp et al., 2013; Cocco et al., 2013).

Specific comments

Line 9 – 10: "natural variability [: : :] are systematically larger than internal variability". This needs to be clearer. I think you mean control estimates are not good enough as they don't include natural external forcings like volcanism, not that model simulated variability is smaller than observed? (or both?)

The sentence has been rephrased to increase the clarity:

However, the natural variability of oxygen ($O_2$) and temperature (T) inferred from the last millennium period is systematically and significantly larger than the internal variability simulated in the corresponding control simulation. This renders estimates of natural variability from control simulations to be biased low

Page 1 Line 12: do you mean "anoxic" or suboxic?

The corresponding text has been removed following the request of reviewer 2 to shorten and streamline the introduction.

Introduction Page 1 line 18. Do you need to reference all these studies to introduce the well-known concept of ventilation age? And then the next statement that warming leads to solubility driven deoxygenation is not referenced.

Again this text has been removed in the revised manuscript.

Page 2 Line 5: Long et al. do not use optimal fingerprinting in their assessment. Please distinguish between studies which use optimal fingerprinting and ToE (these approaches are substantially different since optimal detection studies include an observed change and ToE are primarily model based)

15 We removed this sentence to avoid confusion and for brevity. Andrews et al. (2013) used optimal fingerprinting to detect and attribute changes in marine $O_2$, while Long et al. (2016) evaluate the similarity of the spatial structures associated with natural variability and the forced trend.

20 Page 2 Line 6: For completeness there is another ToE ocean biogeochemistry study by Christian (2014) PLOS ONE

Reference included as suggested.

Page 3 Line 10. As the authors note, multiple studies have done ToE on biogeochemistry for CMIP5-class models. It is necessary in the Introduction to highlight what this study does differently and why it is important. I'm confident this can be done e.g. focus on natural external forcings, millennial scale simulations etc.

As requested by both reviewers, the introduction has been streamlined. We point out what was missing in the literature and what is new in this study. Please see the attached MS with changes highlighted. A few quotes from the introduction are given here below.:

- Studies are missing that address the natural variability of marine $O_2$ and temperature during the recent millennium and that compare this variability with anthropogenic change.

- Studies that address the ToE for temperature in the thermocline are still missing. This is a gap as the observed and projected warming in the thermocline affects the physiology of fish and their habitat distribution in addition to changes in $O_2$ (Pörtner et al., 2014).

- Most of the earlier studies did not consider variability from natural external forcing (e.g. Rodgers et al., 2015; Frölicher et al., 2016; Henson et al., 2017) or only through indirect methods (e.g. Keller et al., 2014; Henson et al., 2016).

Page 3 line 32: here and elsewhere I think the use of the terms like "natural variations in external forcing factors" is confusing. Please use concise, clear terminology e.g. "natural external forcings"

This text has been removed in the revised manuscript, and the terminology "natural variations in external forcing factors" has been replaced by "natural external forcings" as suggested

Page 8. Section 2.2.3: More justification for the chosen standard deviation noise thresholds is required along with reference to the associated statistical confidence levels required.
As requested, the confidence interval has been added:
The threshold of two STD allows the distinction of the signal from the variability with a confidence interval of 95.45 %; this confidence level is selected following many earlier studies (e.g. Christensen et al., 2007).

Figure 1. The different y-axes between left (last millennium) and right (future forcings) and upper (global) and lower (surface) are understandable but confusing given the amount of detail (AOU vs O2sol) in the panel. This Figure needs to be reworked for clarity to focus on key results. E.g. is surface and full depth averaged oxygen concentration important? Or just the thermocline?
The surface and global time series have been moved to the Appendix. Moreover, the thermocline time series has been split into two panels in order to ease the figure: - panel 1 with $O_2$ and T time series and - panel 2 with $O_2$ components time series.

Page 9 – 10 Sect. 3.2. and page 10 – 11 Sect 3.2. These sections are overlong and should be more concise to focus on the key messages.
Sect 3.2 has been condensed.

Figure 4. There is not much added information in this panel – could the Figure be condensed ?
Panel 4a and 4b have been removed.

Figure 5, Sect 3.3.2 and Figure 6, Section 3.3.3: Suggest to (at least) condense these predominantly methodological Sections.
Panels 6a, b, c and d have been removed from the main text. The text has been shortened as requested.
Page 13 Sect. 3.4.1 Lines 8 to 18. I suggest to move this extra analysis of Fig 1 back to Sect 3.1 or simplify Figure 1 and add another Figure here to look at AOU/solubility contributions.
Figure 1 has been simplified as suggested. Panel b) shows now the AOU and $O_{2,sol}$ in the thermocline. We prefer to keep the information on $O_2$, T, and AOU and $O_{2,sol}$ within the same figure for easy comparison. Further, we prefer to keep the discussion on AOU and $O_{2,sol}$ in one single section (Sect. 3.4). Note that section 3.1 to 3.3 discuss $O_2$ and T, but not AOU and $O_{2,sol}$. This clear separation allows us to briefly introduce the concept of AOU and $O_{2,sol}$ at the start of section 3.4.

Page 16 Line 1 - 4. See overall comments on model uncertainty regarding

noise and forced response.
Please see our response to your comment #2.

Section 4. The discussion should be re-written to be more concise and fo-
cused.
The discussion has been shortened and re-structured.

Technical comments

Abstract Line 1. "aggravate" is a little unclear here suggest to reword
Below, the updated sentence:
Marine deoxygenation and anthropogenic ocean warming are observed and pro-
jected to intensify in the future.

Introduction and throughout: many references are used to support each
point. Please be more selective – 13 references for one statement is quite a lot.
The number of citations has been reduced.
Oceanic oxygen $O_2$ concentrations have been observed to decrease over the past
50 years (e.g. Stramma et al., 2008; Helm et al., 2011; Ito et al., 2017; Schmidtko
et al., 2017) and are projected to further decline under anthropogenic climate
change (Sarmiento et al., 1998; Plattner et al., 2001; Keeling and Garcia, 2002;
Shaffer et al., 2009; Cocco et al., 2013; Battaglia and Joos, 2018 ).

Introduction Page 1 line 20. Suggest to cite Capotondi et al. 2012 on strat-
ification
The corresponding text has been removed during the rewrite of the introduction.

Introduction Page 3 line 22. "by appropriately adding [..]" this is unclear
please rework sentence
Below, the updated sentence:
Others used STD from a model ensemble for the 1920-1950 period (Long et al.,
2016) or added the STD of annual values and monthly values in addition to
estimation of measurement uncertainty (Carter et al., 2016).

Introduction Page 3 line line 25 "are considered not at all" rephrase please
This part has been removed.

Introduction Page 3 line 33 "these externally forced variability" rephrase
please
The corresponding sentence has been removed during the rewrite of the intro-
duction.

Introduction Page 3 line 34 repetition of "last millennium climate simula-
tions" rephrase please
Below, the updated sentence:
These include analyses of the last millennium using climate reconstructions

(McGregor et al., 2015; PAGES 2k Consortium et al., 2015; PAGES2k Consortium et al., 2017), climate simulations (Crowley, 2000; Ammann et al., 2007; Fernández-Donado et al., 2013; Camenisch et al., 2016), and the few existing earth system model (ESM) simulations with enabled carbon and biogeochemical cycles (Jungclaus et al., 2010; Lehner et al., 2015; Brovkin et al., 2010; Chikamoto et al., 2016). Regarding biogeochemical cycles, a substantial role of natural forced variability is also found in simulations with and without volcanic forcing (Frölicher et al., 2009; Frölicher et al., 2011; Frölicher et al., 2013).

Introduction Page 4 line 12 remove "the" before 'anthropogenic'
Modified as suggested.

Page 4 line 25: please rephrase (also replace with "rely")
Below, the updated sentence:
This version of the model was used in the Coupled Model Intercomparison Project (CMIP5). Its physics originates from the Community Climate System Model (CCSM4; Gent et al., 2011), which includes modules for the atmosphere, the land, the sea-ice and the ocean, all coupled by a flux coupler.

Page 5 line 3. Replace "stands for"
Below, the updated sentence:
The sea-ice component is the Community Ice Code (CICE4; Hunke et al., 2010). It operates on the same horizontal resolution as the ocean module.

Page 5 line 19. Of carbon?
Below, the update sentence:
The biomass of dead phytoplankton is distributed among dissolved and particulate organic and inorganic carbon and nutrient pools.

Page 6 line 5: Should supplemental figures be named in the order they are introduced? (applies to later sections)
The figure order has been updated.

Page 10. Replace "outweight" with "exceed"
Corrected as suggested.

Page 11 Line 26 "larger"
"typo corrected"

Page 13 – 14. Formatting issues with subscripts
This has been corrected.

Page 15 Line 25 missing word after "enables"
The corresponding sentence has been removed during the rewrite of the introduction

Page 17 Line 2 . Missing parentheses
Corrected.

**2 Reviewer 2**

The authors has presented an interesting study, with a number of valuable analyses and interpretations regarding the Time of Emergence for oxygen and temperature in the ocean. Although the material presented in the manuscript should be of value to the broader research community, it would benefit greatly from revisions to improve clarity, and also to become better anchored in a discussion of existing scientific literature. Suggestions for improving the scientific clarity and impact are detailed below.

MAIN COMMENTS
The Introduction is not sufficiently focused and was a bit all over the map (encyclopedias on multiple topics), and should be streamlined.
We thank the reviewer for his constructive comments. The introduction has been rewritten focusing on the core of the study. We emphasise that we apply the ToE concept to estimate when environmental conditions become unusual compared to the natural variability of the last millennium. We have removed the textbook description on the O2 cycle and shortened and rearranged the text at various places. We refer the editor and reviewers to the attached manuscript where these and other changes are highlighted.

It would be of great value to state clearly why trend detection is priority when interpreting observations, and to relate this to the realm of uncertainty in climate change projections. There are several sources of uncertainty and/or ambiguity in trend detection relating to the "noise" component of trend detection. One issue (that emphasized in the manuscript) is the distinction between natural and internal variability, with there being a need to understand and quantify this distinction. This analysis is great, and warrants emphasis.

But a second issue is related to the way in which noise is calculated. In most studies that also emphasize initial condition large ensemble methods, the method of Deser et al. (2014) is used to estimate noise, with this typically involving linear trends calculated over decades rather than STD of annual means to calculate ToE. As the amplitude of inter annual and decadal variability is typically expected to be distinct, at face value it is not obvious how to connect the estimates give here with more general research using large ensemble simulations. It's very important to emphasize this point, while it seems also OK to point out that there is nothing inherently flawed or wrong with the method proposed in the manuscript, it is just somewhat different, and more similar

to methods that have been applied cross multiple models in inter comparison studies. As a related point, I believe it would be valuable to communicate the implications of this study to the broader community, given the discrepancies noted above.

We have added the following text to point to the work of Deser and colleagues in the introduction:
Studies that employ large model ensembles (Deser et al., 2014) highlight in particular the large contribution of internal natural variability to the spread in projected trends (Rodgers et al., 2015).

We also now explain that the ToE concept is applied for different questions and that we use it to detect "unfamiliar" environmental conditions and not to detect current trends. The corresponding paragraph in the introduction reads now:
The method applied for estimating ToE depends on the scientific questions. Most of the earlier studies on marine $O_2$ are directed towards the detection of the current anthropogenic trend by using modern measurements systems and, closely related, to quantify the uncertainty in projections arising from natural variability (Rodgers et al., 2015; Frölicher et al., 2016; Henson et al., 2016; Long et al., 2016). Studies that employ large model ensembles (Deser et al., 2014) highlight in particular the large contribution of internal natural variability to the spread in projected trends (Rodgers et al., 2015). Alternatively, Henson et al. (2017) address the question when ecosystems are exposed to conditions outside the range of previously experienced seasonal variability and, hence, ToE is estimated relative to preindustrial. In this study, we focus similarly on the detection of persistent unfamiliar conditions. We compare the modelled anthropogenic signal with the natural variability of the entire last millennium for both $O_2$ and temperature in the thermocline.

We provide information in the introduction on large ensemble simulations:
More recently, outputs from a large ensemble of $21^{st}$ century simulations were used to estimate mean trends and the standard deviation in the projected trends in $O_2$ (Rodgers et al., 2015; Frölicher et al., 2016). This approach enables the characterisation of anthropogenically forced trends and the uncertainty in future trends due to internal variability on the same time scale. A drawback is that variability in decadal or, even, multi-decadal trends are difficult to estimate from existing measurements as these cover a short time period. It is therefore difficult to validate the results.

Is there a way to bridge the different methodologies with large ensemble methods, perhaps by sampling (randomly?) decadal trends from a 1000+ year Last Millennium simulation? That would still be different from what is done with large ensemble runs.

This is an interesting idea and may be part of a future study. It is not readily clear to us how the suggested comparison between methodologies should be

done. We do not have large ensemble results for temperature and oxygen avail-
able. Such a comparison appears beyond the scope of this study; we feel that
our manuscript is already quite long, and we prefer not to expand it further by
introducing a new topic.

Do the authors recommend at all major modeling centers that are embarking
on large ensemble simulations also include Last Millennium simulations? Or for
CMIP6 protocols (where the historical period goes through 2014) can potential
biases be estimated by comparing full historical ensemble runs with greenhouse
$CO_2$-only (for example, for estimating internal variability) runs over 1850-2014?
We appreciate this suggestion. We expanded the text in the discussion, section
4.2 to include this point:
Alternatively, large ensemble simulations for the industrial period and the future
will become available within CMIP6. Different ensembles including or exclud-
ing anthropogenic forcing (Stott et al., 2000) and including or excluding natural
forcing may be used to disentangle the individual contributions to trends and
variability. Some model centres may also wish to generate large ensemble simu-
lations for the last millennium to study natural variability over the more recent
preindustrial period (Jungclaus et al., 2010).

MINOR COMMENTS

(1) References Near the top of Pg. 3 the authors refer to Hawkins and Sut-
ton (2012), where the methods considered for calculating Time of Emergence
are not the same as those typically used with large ensemble methods (Deser et
al., 2014). This should be clarified with regard to the comments above.
Done - please see answer to comments above

Also, are the authors sure that the Long et al. (2016) paper used the same
method to calculate noise as that of Hawkins and Sutton?
The corresponding sentence and the reference to Long et al. has been removed
for brevity.

(2) Model configuration is the CAM4 model considered here the same as
the atmospheric model component used by Kay et al. (2015)? More generally,
how do the model components differ, and if so, how might this itself impact
variability?
Kay et al., 2014 use the same version of CESM1.0 as the one used in this study
except for the atmospheric component. They used CESM1.0 with CAM5 where
here we use it with CAM4. CAM5 provides a better aerosol representation
(through physical enhancements), in order to conduct advanced research on
assessment of the aerosol impact on cloud properties. More specifically, it allows
for estimating the impact of anthropogenic aerosol emissions on the radiative
forcing of climate by clouds. The main differences between these two versions
of CAM are described in detail in Liu et al. (2012). Nevertheless, we list below
some of these changes that may influence the variability of the system:

- updated radiation scheme to Rapid Radiative Transfer Method for GCMs (RRTMG). RRTMG has an extensive spectral representation of the water vapour continuum

- the 3-mode modal aerosol scheme (MAM3) has been implemented and provides accumulation and course aerosol modes

- revised cloud macrophysics scheme that imposes full consistency between cloud fraction and cloud condensate

- computation of an updraft vertical velocity which allows for aerosol-cumulus interactions

(3) ToE, Pg. 9, line 8 and ensuing paragraph The section header "ToE" needs to be expanded into "Time of Emergence (ToE)", or something similarly appropriate.
Modified as suggested

It should also be stated explicitly in this paragraph which year is used as a "reference" for the ToE calculations (ToE relative to what year?)
The splined is applied from the year 1800 onward. The corresponding text reads as follows:
The low-frequency climate change, $S$, is diagnosed as the spline-fitted (Enting, 1987) anomalies using a cut-off period of 40 years, from the year 1800 in order to remove short-term variations over the industrial period.

(4) Fig. 2: Pg. 9, describing text The patterns and timescales should be compared with existing published literature for oxygen and temperature, with any caveats about the methods used to calculated noise.
We are not aware of any publication discussing ToE of T in the thermocline. However, ToE of $O_2$ is discussed in several publications. For example, Long et al. (2016) present results from a large ensemble of CESM simulations and Henson et al. (2017) from CMIP5 simulations.
... a rather early emergence is simulated in the eastern equatorial Atlantic and the Indian ocean subtropical gyre. These patterns are consistent with the results from Long et al. (2016) and Henson et al. (2017).

(5) LM, pg 11, noie 25: The authors need to spell out LM as Last Millennium here.
Modified as suggested.

(6) pg 11, lines 30-34 It would improve the clarity of the presentation if a bit more detail were provided here. What are the percent differences, and over what regions?
Information added as requested. The text reads now:
There are many regions where the ratio between the STD from the forced versus the STD from the CTRL simulation is close to one indicating that internal and

total natural variability are approximately equal. In particular, forced and internal natural variability in $O_2$ is comparable in most thermocline regions (Fig. 4a). Indeed, natural variability exceeds internal variability in $O_2$ by more than 50 % in only 3 % of the thermocline, while deviations larger than 20 % are found
5 in 18% of the thermocline. For the temperature, forced variability exceed internal variability by more than 20 % over a third of the thermocline and by more than 50 % over 10 % of the thermocline. The relative difference between natural and internal variability is often smaller for $O_2$ than for T in the thermocline.

10      (7) Section 4.1 on pg 16 The references described here are not appropriate for linking biogeochemistry and climate modes, with the exception of the Bacastow reference. For the case of ENSO it would be appropriate to reference the study of Winguth et al. (1990s?). I believe for the SAM there were the studies in 2006-2007 of LeQuere, Lenton, and Lovenduski, and for the PDO you might
15 consider the study of McKinley (2006).
The suggested references have been added. However, we disagree with the reviewer regarding the references: all studies cited concern the link between biogeochemical variables and a particular climate mode.

20      Interactive comment on Biogeosciences Discuss., https://doi.org/10.5194/bg-2018-523, 2019.

**References**

[revised manuscript text omitted]